solid-state physics/computational physics/software

quantum transport, electronic structure, optical response, disorder, Chebyshev expansions, tight-binding simulations

**Author for correspondence:**
Aires Ferreira
e-mail: aires.ferreira@york.ac.uk

# KITE: high-performance accurate modelling of electronic structure and response functions of large molecules, disordered crystals and heterostructures

Simão M. João[1], Miša Anđelković[2], Lucian Covaci[2], Tatiana G. Rappoport[3,4], João M. V. P. Lopes[1] and Aires Ferreira[5]

[1]Centro de Física das Universidades do Minho e Porto and Departamento de Física e Astronomia, Faculdade de Ciências, Universidade do Porto, 4169-007 Porto, Portugal
[2]Departement Fysica, Universiteit Antwerpen, Groenenborgerlaan 171, 2020 Antwerpen, Belgium
[3]Instituto de Física, Universidade Federal do Rio de Janeiro, Caixa Postal 68528, 21941-972 Rio de Janeiro, Brazil
[4]Centro de Física das Universidades do Minho e Porto and Departamento de Física, Universidade do Minho, Campus de Gualtar, 4710-057 Braga, Portugal
[5]Department of Physics, University of York, York YO10 5DD, UK

MA, 0000-0002-4117-5662; TGR, 0000-0002-1878-5956; AF, 0000-0001-6017-8669

We present KITE, a general purpose open-source tight-binding software for accurate real-space simulations of electronic structure and quantum transport properties of large-scale molecular and condensed systems with tens of billions of atomic orbitals ($N \sim 10^{10}$). KITE's core is written in C++, with a versatile Python-based interface, and is fully optimized for shared memory multi-node CPU architectures, thus scalable, efficient and fast. At the core of KITE is a seamless spectral expansion of lattice Green's functions, which enables large-scale calculations of generic target functions with uniform convergence and fine control over energy resolution. Several functionalities are demonstrated, ranging from simulations of local density of states and photo-emission spectroscopy of disordered materials to large-scale computations of optical conductivity tensors and real-space wave-packet propagation in the presence of magneto-static fields and spin–orbit coupling. On-the-fly calculations of real-space Green's

functions are carried out with an efficient domain decomposition technique, allowing KITE to achieve *nearly ideal linear scaling* in its multi-threading performance. Crystalline defects and disorder, including vacancies, adsorbates and charged impurity centres, can be easily set up with KITE's intuitive interface, paving the way to user-friendly large-scale quantum simulations of equilibrium and non-equilibrium properties of molecules, disordered crystals and heterostructures subject to a variety of perturbations and external conditions.

# 1. Introduction

Computational modelling has become an essential tool in both fundamental and applied research that has propelled the discovery of new materials and their translation into practical applications [1]. The study of condensed phases of matter has benefited from significant advances in electronic structure theory and simulation methodologies. Among these advances are: explicitly correlated wave-function-based techniques achieving sub-chemical accuracy [2], first-principles methods to tackling electronic excitations [3], charge-self-consistent atomistic models for accurate electronic structure calculations [4] and the use of machine learning as means to finding density functionals without solving the Khon–Sham equations [5,6].

Semi-empirical atomistic methods are amongst the simplest and most effective methods to calculate ground- and excited-state properties of materials [7–10]. The increasingly popular tight-binding approach [11] has been employed for accurate and fast calculations of total energies and electronic structure in complex materials, including semiconductors [12,13], quantum dots [14] and superlattices [15,16], and is particularly well-suited for implementation of so-called $O(N)$ algorithms for efficient (linear scaling) calculations of total energies and forces [17]. Because the tight-binding models provide a unified description of electronic bands in crystalline matter, they have been instrumental, for example, in the prediction and classification of topological materials [18–20].

The tight-binding scheme, albeit generally less accurate than *ab initio* methods [21], provides a physically intuitive approach applicable to large systems. By means of a careful parametrization, tight-binding models (obtained by projecting the Hamiltonian onto local orbitals, e.g. maximally localized Wannier functions [22]) allow multi-scale calculations of complex material properties, such as electronic response functions, beyond the scope of first-principles calculations. Accurate tight-binding models have been devised for a plethora of model systems, ranging from metals to ionic materials [23], and shown to correctly predict the optical spectra of multi-shell structures with atomic-scale variations in composition [10], layer-thickness dependence of energy gaps in strained-layer semiconductor superlattices [15], and fine features in the Hofstader butterfly spectrum of moiré graphene superlattices [24].

Tables of energy (hopping) integrals for many elements and crystal structures can be found in the literature (e.g. [25]), while new tight-binding parametrizations can be easily constructed by fitting to experimental data or *ab initio* calculations. Foulkes and Haydock showed that the tight-binding approximation emerges as a stationary solution to density functional theory [26], which has provided a solid foundation to improve the accuracy of tight-binding models. Furthermore, environment-dependent tight-binding models have provided a tool to capture essential features of charge transfer and local environmental dependence of overlap integrals (e.g. to reproduce lattice deformations under strain [27]), which have improved the transferability of the tight-binding approach in a number of model systems, such as silicon and group-III nitrides [23,28–31].

A key feature of the real-space semi-empirical tight-binding (SETB) approach is its versatility. It can accommodate disorder, defects, strain, magnetic interactions and external perturbations as local modifications to the tight-binding parameters, and, as such, provides a powerful framework to tackle realistic non-equilibrium device conditions in nanosystems and mesoscopic structures [32,33]. Recent methodological developments exploiting accurate spectral expansions of lattice Green's functions are rendering an efficient description of ever-larger and complex systems. The underlying principle of the spectral approach—firmly rooted in the Fourier and Chebyshev spectral theory [34]—consists in expanding Green's function in orthogonal polynomials by means of a rapidly converging and stable recursive procedure [24,34,35]. Recent applications include studies of dynamical correlations in the Anderson model [35], phonon lifetimes [36], charge transport [37,38], Monte Carlo simulations [39–41] and matrix product states [42]. Because general-purpose spectral expansions are fully non-perturbative (in some cases even numerically exact [43,44]), they additionally provide a solid benchmark for analytical methods that can be subsequently used to shed light onto the microscopic origin of quantum transport effects.

Some open-source codes already exist and cover different aspects of computational modelling of electronic structure and quantum transport [33]. For example, KWANT is based on the Landauer–Buttiker formalism and the wave function-matching technique for obtaining transport properties from the transmission probabilities of nanodevices which act as scattering regions [45]. PythTB [46] is a Python package with which tight-binding Hamiltonians can be easily defined, and basic quantities, such as the energy dispersion relation or densities of states, can be calculated for small computational domains. Pybinding [47], on the other hand, has a Python interface, and a C++ core, which besides basic diagonalization methods, can also use the kernel polynomial method to model finite size systems with disorder, strains or magnetic fields. GPUQT is a transport code fully implemented for the use on graphical processor units (GPUs), where the size of simulated domains are limited by the device memory to $2 \times 10^7$ [48]. ESSEX-GHOST [49] and TBTK [50] are C++ codes based on the kernel polynomial method that provide Chebyshev expansions of Green's functions for tight-binding models.

Previous numerical implementations of real-space quantum transport, either based on linear response theory or the non-equilibrium Green's function method [51], have so far been limited to mesoscopic structures with up to 10 millions of orbitals [33,37,38,52–55], therefore hampering the extraction of maximum mileage from the tight-binding scheme (except for a recent report, where Kubo calculations with billions of atoms $N = 3.6 \times 10^9$ were demonstrated [43]).

The accessible energy resolution in a tight-binding simulation is fundamentally bounded by the typical mean level spacing of the spectrum, that is, $\eta \gtrsim \delta E$, and hence the system size. At the same time, high-resolution calculations (with $\eta$ of the order of milli-electron-volt (meV) and below) are vital, for example, to capture fine details of multifractal spectra approaching a quantum critical point [56] or to correctly describe coupled charge-spin transport in systems with strong spin–orbit coupling (SOC) [57].

In this environment, KITE strives to bring the tight-binding and spectral expansions methods to the next level in accessible system sizes and energy resolution. It combines the easy set-up of the Python-based packages with a highly optimized and robust C++ core that handles memory-intensive large-scale simulations efficiently. KITE's numerically exact treatment of disordered Green's functions with billions of orbitals enables realistic tight-binding simulations of materials, thus overcoming the difficulties of previous approaches, either based on numerical diagonalization of small systems or in an approximate treatment of disorder effects, such as the single-impurity $T$-matrix, coherent potential approximation [58].

KITE strengths rely on a combination of significant improvements in *versatility, accuracy, speed and scalability*, as summarized in what follows.

*Accuracy and speed*. Target functions are computed employing efficient spectral algorithms based on a numerically exact Chebyshev polynomial expansion of Green's functions discovered independently by Ferreira & Mucciolo [43], A. Ferreira (2014, unpublished) and Braun & Schmitteckert [44]. Within this approach, spectral calculations are carried out with any desired energy resolution, which can be adjusted on-demand by tuning the spectral expansion parameters. KITE exploits the locality of interactions to employ a multi-scale memory management scheme that improves data affinity and dramatically reduces the computation time.

*Versatility*. KITE uses as input arbitrary SETB models in $d = 2$ and 3 spatial dimensions that can be imported from standard formats or defined directly through its user-friendly Python interface. A real-space 'disorder cell' approach is used to set up on-site/bond disorder and defects, such as vacancies, random gauge fields or spin-dependent disorder. Model and calculation parameters can be modified without the need to recompile the core C++ code.

*Methodology and implementation*. A large-RAM 'single-shot' recursive algorithm is used to evaluate Fermi surface properties of large-scale systems with multi-billions of orbitals $N \sim 10^{10}$ (A. Ferreira 2014, unpublished). In SETB models with small coordination number $[Z = O(1)]$, the high-resolution numerical evaluation of $T = 0$ DC conductivity (incorporating tens of thousands of spectral moments for individual $N \times N$ Green's functions) takes only a few hours with 16 cores [43]. This gives access to accuracy and energy resolutions up to three orders of magnitude beyond previous approaches [24,32,33]. To assess response tensors $\hat{\sigma}(\omega, T)$ at finite temperature/frequency, KITE implements its accurate spectral approach to carry out a direct evaluation of the Kubo–Bastin formula following Garcia, Covaci and Rappoport in [53]. A similar approach is employed to evaluate nonlinear optical conductivities of non-interacting systems [59].

*Functionalities*. The KITE software package has the following functionalities:

— average density of states (DOS) and local DOS;
— spectral function;

— generic multi-orbital local (on-site) and bond disorder;
— linear response DC conductivity tensors;
— linear and nonlinear optical (AC) conductivity tensors; and
— wave-packet propagation,

for generic SETB models, in the presence of magnetic fields and disorder.

*Scalability.* Quantum transport calculations in disordered conductors and complex structures with large unit cells (e.g. twisted bilayers of van der Waals materials) require simulations with very large $N$. To optimize multi-threading and speed-up spectral expansions, KITE provides the option to thread pre-defined partitions in real space through a domain decomposition technique in close resemblance to the ones used in specialized pseudospectral and spectral techniques for solving partial differential equations [34,60].

The remainder of the paper is organized as follows. The theoretical and conceptual foundations of KITE are presented in §2, the description of the code and its organization is presented in §3, examples illustrating KITE's versatility are given in §4, while the performance of the code is benchmarked in §5. Finally, §6 presents our conclusions, outlook and possible future developments.

# 2. Basic theoretical concepts

The SETB method is a computationally fast and robust approach to handle large-scale molecular and condensed matter systems [23]. In the tight-binding approximation, electrons are assumed to be strongly bound to the nuclei. One-particle wave functions $\{\psi_\alpha(\mathbf{x})\}$ are approximated by linear combinations of Slater–Koster-type states for isolated atoms, i.e.

$$\psi_\alpha(\mathbf{x}) = \frac{1}{\sqrt{N}} \sum_{i=1}^{N} a_\alpha(i) \phi_{\mathrm{SK}}(\mathbf{x} - \mathbf{x}_i),\tag{2.1}$$

where $i = \{1 \cdots N\}$ runs over all sites and orbitals. The one-particle states $|\psi_\alpha\rangle$ are eigenvectors of the parametrized Hamiltonian matrix $\hat{\mathcal{H}} = \sum_{i,j} t_{i,j} |i\rangle\langle j|$. SETB matrix elements—encoding on-site energies ($i = j$) and hopping integrals between different atomic orbitals ($i \neq j$)—can be estimated by means of the two-centre Slater–Koster formulation or by fitting to experimental data or first-principles calculations for suitable reference systems [11,61,62]. For example, SETB models can be derived using Tight-Binding Studio [63], which provides Slater–Koster coefficients with orthogonal or non-orthogonal basis sets.

Crucially, the SETB complexity grows only linearly with the number of atomic orbitals, thus enabling large-scale calculations of a plethora of equilibrium and non-equilibrium physical properties, including optical absorption spectra and electronic response functions, simulations of amorphous solids and wave-packet propagation. Disorder, interfaces and defects can be conveniently added to an SETB model by modifying on-site energies and hopping integrals and adding auxiliary sites. Such a multi-scale approach has proven useful in describing impurity scattering [38,64], moiré patterns [55,65], complex interactions induced by adatoms [66], disorder-induced topological phases [67], optical conductivity of two-dimensional materials with up to tens of millions of atoms [54] and geometrical properties, vibrational frequencies and interactions of large molecular systems [68].

## 2.1. Spectral methods: a crash course

The electronic properties of molecular and condensed systems are encoded in the eigenvalues and eigenvectors of Hamiltonian matrices $\hat{\mathcal{H}}$ with large dimension $N$. Evaluation of spectral properties and correlation functions via numerically exact diagonalization requires memory of the order of $N^2$, while the number of floating-point operations scale as $N^3$. Such a large resource consumption restricts the type and size of systems that can be handled by direct diagonalization. Spectral methods offer a powerful and increasingly popular alternative. In the spectral approach, the target function of interest is decomposed into a spectral series

$$f(E) \propto \sum_{n=0}^{\infty} f_n P_n(E),\tag{2.2}$$

where $\{P_n(E)\}_{n \geq 0}$ is an orthogonal polynomial sequence. The elegance and usefulness of the spectral

decomposition lie in the fact that the expansion moments $f_n$ can be retrieved to any desired accuracy by means of a highly-efficient and stable recursive scheme. The spectral expansion equation (2.2) is guaranteed to converge (in the usual (norm) sense) provided that the target interval is free of singularities [34]. The family of Chebyshev polynomials of the first kind

$$T_0(x) = 1, \tag{2.3}$$

$$T_1(x) = x \tag{2.4}$$

and
$$T_{n+1}(x) = 2xT_n(x) - T_{n-1}(x), \tag{2.5}$$

with $x \in \mathcal{L} \equiv [-1:1]$, provides a robust general-purpose basis function set thanks to its unique spectral convergence properties and intimate relation to the Fourier transform [34]. The Chebyshev polynomials $\{T_n(\arccos x) = \cos(nx)\}$ satisfy the orthogonality relations

$$\int_{\mathcal{L}} \mathrm{d}x\, \omega(x) T_n(x) T_m(x) = \frac{1 + \delta_{n,0}}{2} \delta_{m,n}, \tag{2.6}$$

with $\omega(x) = 1/(\pi\sqrt{1 - x^2})$ and thus form a complete set on $\mathcal{L}$. With this choice of basis functions, the spectral decomposition can be written as

$$f(x) = \omega(x) \sum_{n=0}^{\infty} \frac{1 + \delta_{n,0}}{2} \mu_n\, T_n(x), \tag{2.7}$$

where $\mu_n$ are the so-called *Chebyshev moments*

$$\mu_n = \int_{\mathcal{L}} \mathrm{d}x\, f(x) T_n(x). \tag{2.8}$$

The extension of equation (2.7) to operators $f(\hat{X})$ is straightforward. The efficient implementation of matrix polynomial expansions allows one to tackle complex quantum-mechanical problems bypassing direct diagonalization. The operators $\mathcal{T}_n(\hat{X})$ are constructed using the matrix version of the standard Chebyshev recursion relations

$$\mathcal{T}_0(\hat{X}) = 1, \tag{2.9}$$

$$\mathcal{T}_1(\hat{X}) = \hat{X} \tag{2.10}$$

and
$$\mathcal{T}_{n+1}(\hat{X}) = 2\hat{X}\mathcal{T}_n(\hat{X}) - \mathcal{T}_{n-1}(\hat{X}), \tag{2.11}$$

where $\hat{X}$ is any square matrix with eigenvalues in the canonical interval ($\|\hat{X}\| \leq 1$). Typical target functions include the exponential operator, Dirac deltas and Green's functions. For example, the Chebyshev expansion of the familiar 'spectral operator' $\delta(\epsilon - \hat{H})$ is given by [35]

$$\delta(\epsilon - \hat{H}) = \sum_{n=0}^{\infty} \Delta_n(\epsilon) \frac{\mathcal{T}_n(\hat{H})}{1 + \delta_{n,0}}, \tag{2.12}$$

where

$$\Delta_n(\epsilon) = \frac{2\, T_n(\epsilon)}{\pi\sqrt{1 - \epsilon^2}}. \tag{2.13}$$

The rescaled (dimensionless) $\hat{H}$ and $\epsilon$ are obtained from the original Hamiltonian $\hat{\mathcal{H}}$ and energy variable $E$ via a simple linear transformation

$$\hat{H} = \frac{\hat{\mathcal{H}} - \delta\epsilon_+}{\delta\epsilon_-} \tag{2.14}$$

and

$$\epsilon = \frac{E - \delta\epsilon_+}{\delta\epsilon_-}, \tag{2.15}$$

where $\delta\varepsilon_{\pm} = (E_{\max} \pm E_{\min})/2$ and $\varepsilon_{\max(\min)}$ are the upper (lower) endpoints of the spectrum.

The spectral decomposition (equation (2.12)) provides the starting point for an accurate and efficient calculation of several important quantities, for example, the average DOS

$$\rho(\epsilon) = \frac{1}{N} \mathrm{Tr}\, \delta(\epsilon - \hat{H}) = \frac{1}{\pi\sqrt{1 - \epsilon^2}} \sum_{n=0}^{\infty} \mu_n T_n(\epsilon), \tag{2.16}$$

where $N = \dim \hat{H}$ is the Hilbert space dimension. The Chebyshev moments,

$$\mu_n = \frac{1}{N}\frac{(1 + \delta_{n,0})}{2}\mathrm{Tr}\, \mathcal{T}_n(\hat{H}) \tag{2.17}$$

are evaluated recursively in two steps. First, a series of matrix-vector multiplications are carried out to construct the Chebyshev matrix polynomials using equation (2.9). The complexity for sparse SETB matrices is $Z \times N$ (per Chebyshev iteration), where $Z$ is lattice coordination number. The trace is evaluated using a stochastic technique (see below). Finally, the DOS is reconstructed over a grid of energies with a number $M$ of calculated Chebyshev moments, yielding the $M$th-order approximation to the DOS

$$\rho(\epsilon) \simeq \frac{1}{\pi\sqrt{1 - \epsilon^2}}\sum_{n=0}^{M-1}\mu_n T_n(\epsilon). \tag{2.18}$$

Chebyshev expansions provide uniform resolution due to errors being distributed uniformly on $\mathcal{L}$ [34]. In principle, the spectral resolution is only limited by the number of moments retained in the expansion. As a rule of thumb, the resolution is inversely proportional to the number of moments used, $\delta E \sim \Delta E/M$, where $M - 1$ is the highest polynomial order and $\Delta E$ is the spectrum bandwidth (prior to re-scaling). In specific problems, the scaling with the number of moments can be substantially more demanding (e.g. near a singularity in the DOS [24,43]).

Truncated spectral expansions, such as equation (2.18), can exhibit spurious features known as Gibbs oscillations caused by discontinuities or singularities (a familiar case is the 'ringing' artefact appearing in the Fourier expansion of a square wave signal). Gibbs oscillations can be cured using specialized filtering techniques. A popular approach in quantum chemistry is the so-called kernel polynomial method (KPM) [35]. As the name suggests, the KPM makes use of convolutions with a kernel to attenuate the Gibbs oscillations, e.g. for the DOS, $\mu_n \rightarrow \mu_n \times g_n$. A popular choice is the so-called Lorentz kernel, $g_n^L = \sinh(\lambda(1 - n/M))/\sinh(\lambda)$, where $\lambda$ is an adjustable resolution parameter. This kernel has the property that approximates nascent Dirac-delta functions $\delta_\eta(x)$ by a Lorentzian with resolution $\eta = \lambda/M$, and thus has been employed to damp Gibbs oscillations in the spectral decomposition of lattice Green's functions [35]. A powerful alternative is given by the Chebyshev polynomial Green's function (CPGF) method [43] (A. Ferreira 2014, unpublished), which is based on the exact spectral decomposition of the resolvent operator

$$g^{\sigma,\eta}(\epsilon, \hat{H}) = \frac{\hbar}{\epsilon - \hat{H} + i\sigma\eta} = \hbar \sum_{n=0}^{\infty} g_n^{\sigma,\eta}(\epsilon)\frac{\mathcal{T}_n(\hat{H})}{1 + \delta_{n,0}}, \tag{2.19}$$

where

$$g_n^{\sigma,\eta}(\epsilon) = -2\sigma i \frac{e^{-\sigma i n \arccos(\epsilon + i\sigma\eta)}}{\sqrt{1 - (\epsilon + i\sigma\eta)^2}}, \tag{2.20}$$

derived in [43,44]. The expansion coefficients $g^{\sigma,\eta}$ (equation (2.20)) encompass retarded ($\sigma = +$) and advanced ($\sigma = -$) sectors, with $\eta > 0$ by definition. In contrast to the Lorentz kernel, these exact spectral coefficients depend explicitly on the energy. The CPGF expansion (equation (2.12)) is separated into a polynomial function of $\hat{H}$ and a coefficient which encapsulates the frequency and energy parameters, similar to the delta-function decomposition in equation (2.12). This implies that the real-space Green's function (and any related target function) can be quickly evaluated for any $\epsilon$, $\eta$ once the Chebyshev moments have been determined. Moreover, asymptotic convergence of the $M$th-order approximation to Green's function is guaranteed to any desired accuracy. KITE combines the advantageous properties of the CPGF expansion with an efficient (linear-scaling) domain decomposition technique (§3.2) to enable accurate calculations of real-space electronic structure and quantum transport properties in unprecedented large SETB models.

Figure 1$a$ illustrates the truncated KPM and CPGF expansions for the Lorentzian function $f_L(x) = \eta/(\pi(x^2 + \eta^2))$, which serves as a proxy to infer the convergence properties of the spectral method in an actual Green's function calculation. The numerically exact CPGF expansion manifestly converges for any value of $x$ (figure 1$b$). In contrast, the KPM approximation leads to a small overshooting (this effect is exaggerated in Fig. 1$a$ for clarity). The KPM error is negligible for most practical purposes, except close to the interval ends, where its expansion coefficients formally diverge (see Eq. 2.13). Figure 1$d$ shows the convergence of the DOS at the band centre $E = 0$ eV of a giant honeycomb lattice with 3.6 billion sites with a dilute concentration of random vacancies. Clearly, the CPGF method converges faster than the KPM for all values of $\eta$. Away from the band centre, only

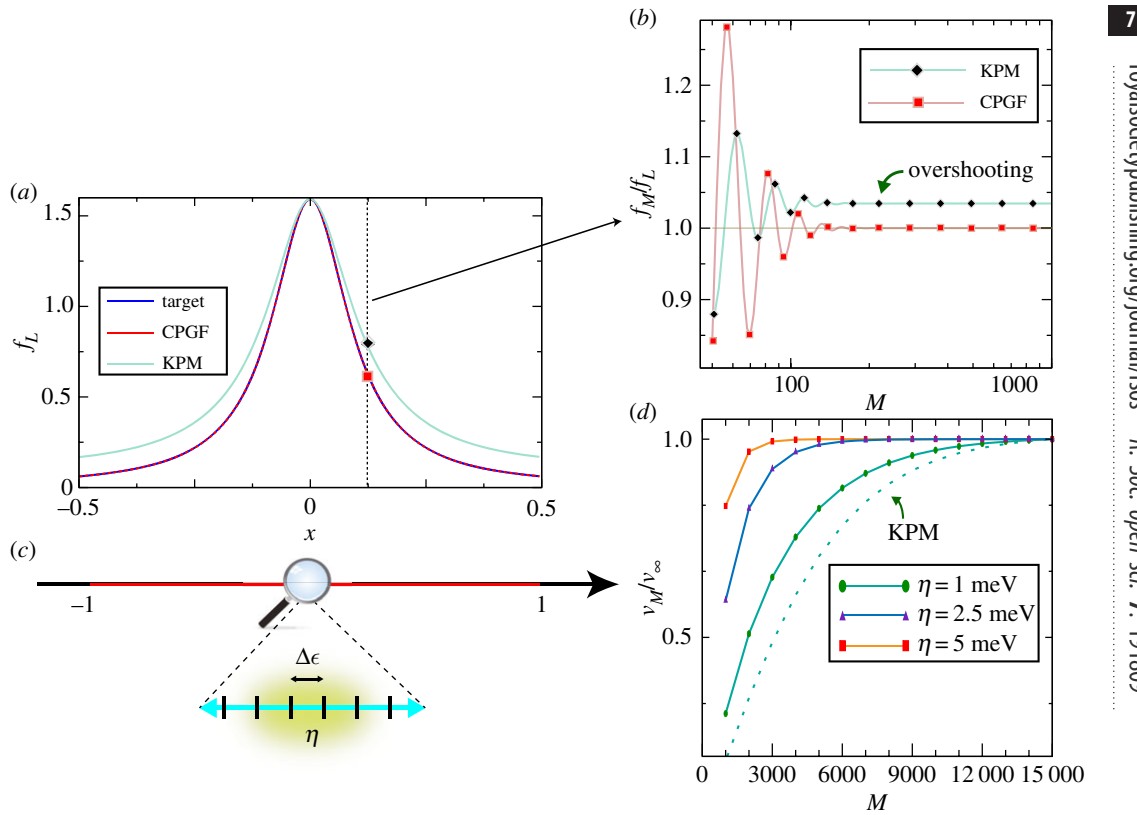

**Figure 1.** (a) Approximation to the Lorentzian curve $f_L(x)$ using the KPM with a Lorentz kernel [35] and the numerically exact CPGF method [43] (A. Ferreira 2014, unpublished). (b) Spectral convergence of the $M$th-order approximation to the target function. For large $M$, KPM converges to $f_M/f_L > 1$ (overshooting), while the CPGF is asymptotically exact $f_M/f_L \to 1$. (c) Energy resolution (broadening) $\eta$ and energy-level separation $\Delta\varepsilon$ for simulations in a finite system. (d) Convergence of $M$th-order approximation to the DOS at the band centre of a giant honeycomb lattice with $60\,000 \times 60\,000$ sites and vacancy defect concentration of 0.4%. As a guide to the eye, the ratio of the DOS normalized to its converged value (to 0.1% accuracy) is plotted. The DOS obtained from a KPM expansion with a Lorentz kernel is shown for $\eta = 1$ meV. Panel (d) is adapted from Ferreira & Mucciolo [43].

the CPGF asymptotically converges to the true (thermodynamic) value. We note in passing that in spectral calculations of this type, $\eta$ needs to be larger than the discrete energy-level separation of the finite system to avoid spurious results. The thermodynamic DOS is obtained by letting $N \to \infty$ (and thus $\Delta\varepsilon \to 0$) prior to $\eta \to 0$ (figure 1c). For additional discussions, the reader is referred to supplemental information in [43].

## 2.2. Linear and nonlinear conductivity tensors

In what follows, we present the scheme for calculation of electronic response functions. For an SETB model subjected to an external electric field $E(t) = -\partial_t A(t)$, the current operator is calculated directly from the Hamiltonian using $\hat{J}^{\alpha} = -\Omega^{-1} \partial H / \partial A^{\alpha}$ ($\Omega$ is the volume and $\alpha = x, y, z$ labels the spatial direction),

$$\hat{J}^{\alpha}(t) = -\frac{e}{\Omega}\left(\hat{h}^{\alpha} + e\hat{h}^{\alpha\beta}A^{\beta}(t) + \frac{e^2}{2!}\hat{h}^{\alpha\beta\gamma}A^{\beta}(t)A^{\gamma}(t) + \cdots\right), \tag{2.21}$$

where we have defined

$$\hat{h}^{\alpha_1\cdots\alpha_n} = \frac{1}{(i\hbar)^n}[\hat{r}^{\alpha_1}, [\cdots[\hat{r}^{\alpha_n}, \hat{H}]]], \tag{2.22}$$

with $\hat{r}$ being the position operator. To first order, $\hat{h}^{\alpha}$ is just the single-particle velocity operator, and the

conductivity tensor can be written as

$$\sigma^{\alpha\beta}(\omega) = \frac{ie^2}{\Omega\omega}\int_{-\infty}^{\infty} d\epsilon f(\epsilon) \text{Tr}\left[\hat{h}^{\alpha\beta}\delta(\epsilon - \hat{H}) + \frac{1}{\hbar}\hat{h}^{\alpha}g^R(\epsilon + \hbar\omega)\hat{h}^{\beta}\delta(\epsilon - \hat{H})\right.$$
$$\left. + \frac{1}{\hbar}\hat{h}^{\alpha}\delta(\epsilon - \hat{H})\hat{h}^{\beta}g^A(\epsilon - \hbar\omega)\right]. \tag{2.23}$$

Similarly, the non-symmetrized second-order conductivity becomes

$$\sigma^{\alpha\beta\gamma}(\omega_1, \omega_2) = \frac{1}{\Omega}\frac{e^3}{\omega_1\omega_2}\int_{-\infty}^{\infty} d\epsilon f(\epsilon) \text{Tr}\left[\frac{1}{2}\hat{h}^{\alpha\beta\gamma}\delta(\epsilon - \hat{H})\right.$$
$$+ \frac{1}{\hbar}\hat{h}^{\alpha\beta}g^R(\epsilon + \hbar\omega_2)\hat{h}^{\gamma}\delta(\epsilon - \hat{H})$$
$$+ \frac{1}{\hbar}\hat{h}^{\alpha\beta}\delta(\epsilon - \hat{H})\hat{h}^{\gamma}g^A(\epsilon - \hbar\omega_2)$$
$$+ \frac{1}{2\hbar}\hat{h}^{\alpha}g^R(\epsilon + \hbar\omega_1 + \hbar\omega_2)\hat{h}^{\beta\gamma}\delta(\epsilon - \hat{H})$$
$$+ \frac{1}{2\hbar}\hat{h}^{\alpha}\delta(\epsilon - \hat{H})\hat{h}^{\beta\gamma}g^A(\epsilon - \hbar\omega_1 - \hbar\omega_2)$$
$$+ \frac{1}{\hbar^2}\hat{h}^{\alpha}g^R(\epsilon + \hbar\omega_1 + \hbar\omega_2)\hat{h}^{\beta}g^R(\epsilon + \hbar\omega_2)\hat{h}^{\gamma}\delta(\epsilon - \hat{H})$$
$$+ \frac{1}{\hbar^2}\hat{h}^{\alpha}g^R(\epsilon + \hbar\omega_1)\hat{h}^{\beta}\delta(\epsilon - \hat{H})\hat{h}^{\gamma}g^A(\epsilon - \hbar\omega_2)$$
$$\left. + \frac{1}{\hbar^2}\hat{h}^{\alpha}\delta(\epsilon - \hat{H})\hat{h}^{\beta}g^A(\epsilon - \hbar\omega_1)\hat{h}^{\gamma}g^A(\epsilon - \hbar\omega_1 - \hbar\omega_2)\right]. \tag{2.24}$$

The task at hand is to compute efficiently the traces of operators containing combinations of the retarded and advanced Green's functions, Dirac-delta and the generalized velocity operators, computed with the CPGF expansion, equations (2.19) and (2.20).

The trace in the conductivity becomes a trace over a product of Chebyshev polynomials and $\hat{h}$ operators, which is encapsulated in the $\Gamma$ tensor

$$\Gamma_{n_1\cdots n_m}^{\boldsymbol{\alpha}_1, \ldots, \boldsymbol{\alpha}_m} = \frac{\text{Tr}}{N}\left[\tilde{h}^{\boldsymbol{\alpha}_1}\frac{T_{n_1}(\hat{H})}{1 + \delta_{n_1,0}}\cdots\tilde{h}^{\boldsymbol{\alpha}_m}\frac{T_{n_m}(\hat{H})}{1 + \delta_{n_m,0}}\right]. \tag{2.25}$$

The upper indices in bold stand for any number of indices (e.g. $\boldsymbol{\alpha}_1 = \alpha_1^1\alpha_1^2\cdots\alpha_1^{N_1}$). KITE uses $\tilde{h}^{\boldsymbol{\alpha}_1} = (i\hbar)^{N_1}\hat{h}^{\boldsymbol{\alpha}_1}$ to avoid handling complex numbers for real SETB models. It is important to notice that these new operators are no longer Hermitian. The commas in $\Gamma$ separate the various $\tilde{h}$ operators. $N$ is the dimension of the Hilbert space and ensures that $\Gamma$ is an intensive quantity. Some examples of possible $\Gamma$ matrices with different complexities are given below

$$\Gamma_{nm}^{\alpha,\beta\gamma} = \frac{\text{Tr}}{N}\left[\tilde{h}^{\alpha}\frac{\mathcal{T}_n(\hat{H})}{1 + \delta_{n,0}}\tilde{h}^{\beta\gamma}\frac{\mathcal{T}_m(\hat{H})}{1 + \delta_{m,0}}\right], \tag{2.26}$$

$$\Gamma_n^{\alpha\beta} = \frac{\text{Tr}}{N}\left[\tilde{h}^{\alpha\beta}\frac{\mathcal{T}_n(\hat{H})}{1 + \delta_{n,0}}\right] \tag{2.27}$$

and

$$\Gamma_{nmp}^{\alpha,\beta,\gamma} = \frac{\text{Tr}}{N}\left[\tilde{h}^{\alpha}\frac{\mathcal{T}_n(\hat{H})}{1 + \delta_{n,0}}\tilde{h}^{\beta}\frac{\mathcal{T}_m(\hat{H})}{1 + \delta_{m,0}}\tilde{h}^{\gamma}\frac{\mathcal{T}_p(\hat{H})}{1 + \delta_{p,0}}\right]. \tag{2.28}$$

The CPGF coefficients can also be cast into a matrix, which we call $\Lambda$. Examples of possible coefficients associated with the previous $\Gamma$ matrices are given below:

$$\Lambda_n = \int_{-\infty}^{\infty} d\epsilon f(\epsilon)\Delta_n(\epsilon), \tag{2.29}$$

$$\Lambda_{nm}(\omega) = \hbar\int_{-\infty}^{\infty} d\epsilon f(\epsilon)\left[g_n^R(\epsilon + \hbar\omega)\Delta_m(\epsilon) + \Delta_n(\epsilon)g_m^A(\epsilon - \hbar\omega)\right], \tag{2.30}$$

and

$$\Lambda_{nmp}(\omega_1, \omega_2) = \hbar^2 \int_{-\infty}^{\infty} d\epsilon\, f(\epsilon)\, [g_n^R(\epsilon + \hbar\omega_1 + \hbar\omega_2)g_m^R(\epsilon + \hbar\omega_2)\Delta_p(\epsilon)$$
$$+ g_n^R(\epsilon + \hbar\omega_1)\Delta_m(\epsilon)g_p^A(\epsilon - \hbar\omega_2)$$
$$+ \Delta_n(\epsilon)g_m^A(\epsilon - \hbar\omega_1)g_p^A(\epsilon - \hbar\omega_1 - \hbar\omega_2)]. \quad (2.31)$$

In terms of these new objects, the conductivities become

$$\sigma^{\alpha\beta}(\omega) = \frac{-ie^2}{\Omega_c\hbar^2\omega}\left[\sum_n \Gamma_n^{\alpha\beta}\Lambda_n + \sum_{nm} \Lambda_{nm}(\omega)\Gamma_{nm}^{\alpha,\beta}\right], \quad (2.32)$$

in first order and

$$\sigma^{\alpha\beta\gamma}(\omega_1, \omega_2) = \frac{ie^3}{\Omega_c\omega_1\omega_2\hbar^3}\left[\frac{1}{2}\sum_n \Lambda_n\Gamma_n^{\alpha\beta\gamma} + \sum_{nm}\Lambda_{nm}(\omega_2)\Gamma_{nm}^{\alpha\beta,\gamma}\right.$$
$$\left. + \frac{1}{2}\sum_{nm}\Lambda_{nm}(\omega_1 + \omega_2)\Gamma_{nm}^{\alpha,\beta\gamma} + \sum_{nmp}\Lambda_{nmp}(\omega_1, \omega_2)\Gamma_{nmp}^{\alpha,\beta,\gamma}\right], \quad (2.33)$$

in second order. $\Omega_c$ is the volume of the unit cell. The derivation of the spectral expansion of linear and nonlinear conductivity tensors can be found in [58]. The convergence of the spectral series is discussed in [43,53].

### 2.2.1. Numerical storage of $\Gamma$ and memory usage

Each entry in a $\Gamma$ matrix is (in general) a complex number, which is represented by two double-precision floating-point numbers, each taking up 8 bytes of storage. Thus, the memory needed to store a $\Gamma$ matrix of rank $n$ is $16\,M^n$. The number of Chebyshev polynomials required for a given resolution depends on the model. As an example, for $M = 1024$, a $\Gamma$ matrix of rank 1 would take up 16 kB of storage, a rank 2 matrix 16 MB and a rank 3 matrix 16 GB. Rank 3 matrices appear in the second-order conductivity.

## 2.3. Large-scale tight-binding simulations

The required number of moments $M$ can drastically increase as electronic states are probed with finer energy resolutions. This becomes especially challenging when evaluating response functions that are products of Green's functions, as in the case of $T = 0$ longitudinal conductivity [38] or at finite temperature/frequency, where off-Fermi surface processes are relevant [53]. To overcome this difficulty, KITE combines different numerical strategies, including a 'single-shot' algorithm for fast evaluation of Fermi surface contributions to electronic response functions. This algorithm bypasses the expensive double recursive calculation of moments (equation (2.26)), allowing to tackle demanding quantum transport problems [43].

In order to optimize the evaluation of the trace operation $\text{Tr}\{\mathcal{T}_n(H)\cdots\mathcal{T}_m(H)\}$, KITE uses the stochastic trace evaluation technique (STE) for the evaluation of target functions requiring a trace over the complete Hilbert space, such as the average DOS and DC conductivity. For example, for the average DOS, the STE trace reads

$$\rho_{\text{STE}}(\epsilon) = \frac{1}{R}\sum_{r=1}^{R}\langle r|\delta(\epsilon - \hat{H})|r\rangle, \quad (2.34)$$

with random vectors $|r\rangle = \sum_{i=1}^N \xi_{r,i}|i\rangle$. The random variables $\xi_{r,i}$ are real- or complex-valued and fulfil white-noise statistics: $\langle\langle\xi_{r,i}\rangle\rangle = 0$, $\langle\langle\xi_{r,i}\xi_{r',i'}\rangle\rangle = 0$ and $\langle\langle\xi_{r,i}^\star\xi_{r',i'}\rangle\rangle = \delta_{r,r'}\delta_{i,i'}$ [35].

The STE is exceptionally accurate for large sparse matrices (only a few random vectors are needed to converge to many decimal places), which allows substantial savings in computational time. For example, the evaluation of Chebyshev moments of the DOS function requires a total number of operations scaling as

$$P_{\text{DOS}} = Z \times N \times M \times R \times S, \quad (2.35)$$

where $S$ is the number of disorder realizations (for clean systems $S = 1$). The required number of random

vectors $R$ depends on the sparsity degree of $\mathcal{T}_n(H)$ [43]. For self-averaging calculations with a very large number of orbitals ($N \gg 1$, $S = O(1)$), a single random vector often suffices to achieve accuracy of 1% or better. However, strictly speaking, only for very sparse matrices, the STE relative error has the favourable scaling often alluded to in the literature, i.e. $1/\sqrt{RN}$ [35], which is typically achieved only for small $M$ and $Z$. Moreover, the number of moments $M$ required to converge the spectral expansion (to a given desired accuracy) depends sensitively on the target resolution, and thus $M$ in equation (2.35) is effectively a function of $\eta$. All together, numerical calculations of average DOS have an intrinsic complexity

$$P_{\text{DOS}} \sim N * f(N, \eta), \tag{2.36}$$

where the function $f(N, \eta) \sim N^{\alpha(\eta)}$ has a favourable polynomial scaling with $N$ for most problems ($1 > \alpha(\eta) > 1/2$) [43] (A. Ferreira 2014, unpublished). For this reason, the spectral approach is substantially less demanding than direct diagonalization techniques even in problems requiring fine resolution.

# 3. Organization and general description of the KITE code

KITE is designed and optimized to maximize accessible system sizes, to improve self-averaging and to achieve fine energy resolution. SETB Hamiltonian matrices have typically $O (Z \times N)$ non-zero entries. The elementary matrix-vector multiplication, at the heart of the Chebyshev iteration scheme, could be done with any standard sparse matrix scheme if all the non-zero entries of the Hamiltonian are stored [49]. Depending on the lattice, the storage of the Hamiltonian can entail a relevant computational cost. Using graphene as an example, if one considers the first and second neighbour hoppings, the number of non-zero elements of the Hamiltonian is 18$N$, which is much larger than the 2$N$ elements needed to store two vectors for the CPGF recursion. KITE avoids storage of the periodic part of the Hamiltonian by using a set of pattern rules to encode it.

Within this section we review the design options that rule KITE's development in order to maximize its efficiency and flexibility.

## 3.1. Template design of the operators

A tight-binding model is constructed with the translation of unit cells by lattice vectors. The hoppings are encoded by a floating-point number with the value of the hopping integral and an integer associated with the distance between orbitals. For a periodic system, these numbers are kept constant along the lattice, which permits encoding $N$ elements into just two numbers. This property leads to massive memory saving and efficient vector/matrix multiplication.

For structural defects and impurities, a similar strategy based on patterns is implemented. A defect is a set of local energies and hoppings connecting orbitals in the vicinity of a lattice site. A set of integers with the positions of the (random) reference lattice point are stored in memory. The memory usage is proportional to the concentration $c$. Similar to Anderson on-site disorder, defects entail a memory cost of the order of $\kappa N$ with $\kappa \ll 2$, which is typically much smaller then the memory necessary to store the vectors required for the Chebyshev iteration.

## 3.2. Efficiency and parallelization

Spectral methods are memory-bound due to their low arithmetic intensity. Minimizing redundant information reduces memory transfers during computations, generating a positive impact on performance. KITE exploits the locality of the Hamiltonian in real space to improve memory management through a multi-scale approach, depicted in figure 2.

On the large scale, KITE uses a domain decomposition strategy to distribute spatial regions through the available processors, designed to improve data affinity. Neighbouring domains are not fully independent, and thus each recursive iteration requires communication between processors. By adding a 'ghost' to each subdomain, which is an extra layer at the borders with copies of elements of the neighbour subdomains, it can independently perform the full iteration. After the iteration, it synchronizes the ghosts between all subdomains. The ghosts synchronization is the non-parallel component of the algorithm, leading to an efficiency bottleneck. Fortunately, this bottleneck scales with the border/volume ratio, which is negligible for large system sizes.

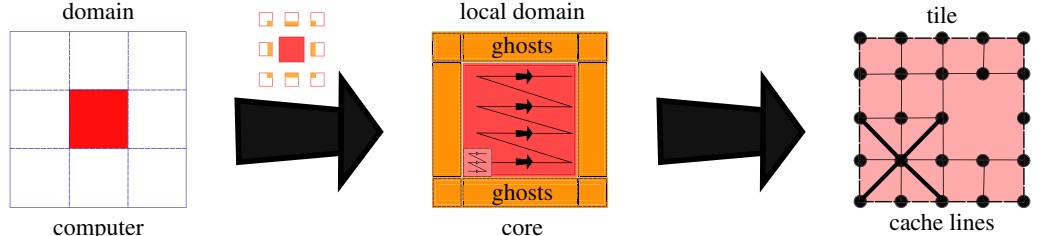

**Figure 2.** Multi-scale organization of KITE. The lattice (left) is divided into domains, which are assigned to different computing processor units. Each core also gets information about the neighbouring domains (orange slices in the middle image). Each domain is divided into TILES and the computation in each core is done inside each of these tiles first, before moving on to a neighbouring tile. Finally, each tile is composed of several unit cells (right).

At the subdomain level, KITE has a smaller length scale, a TILE, to reorganize the matrix-vector multiplication. The subdomain is tiled by identical $D$-dimensional hypercubic sections of linear length TILE. The multiplication is then performed inside each hypercube before moving on to the next one. This reorganization of the multiplication permits two significant optimizations:

— independently of the number of dimensions, TILE is always defined on compilation, and it controls the size of memory chunks. This allows the vectorization of the inner loop in the matrix/vector multiplication;
— contributions for each vector element are determined by the neighbours in all directions. Multiplications along the memory alignment could result in a memory element being called each time the multiplication is performed on one of its neighbours, as the lattice typically exceeds the cache memory. Iterating sequentially inside the small hypercube, allows KITE to fully fit it inside the memory cache and minimize the transfers and cache misses.

During the regular multiplication (the one pertaining to the periodic part of the Hamiltonian) inside each of the hypercubes, KITE is also performing the multiplication related to the disorder and defects, leading to a major performance boost. The ideal value of this TILE compilation parameter is highly dependent on the hardware architecture and should be optimized by the user to allow maximal performance.

## 3.3. KITE workflow

The KITE workflow is divided into three phases (figure 3). First, the user specifies the SETB model on a Python configuration script by using the Pybinding syntax. This configuration file also includes information about the target functions to be evaluated, number of energy points required, etc. The model, together with the calculation settings, is exported to a HDF5 file which becomes the I/O of the main program (KITEx).

In the second phase, the pre-compiled KITEx executable reads the HDF5 file and computes the matrices of Chebyshev polynomials that correspond to each of the requested quantities (DOS, optical conductivity, etc). This is the most demanding step of the calculation. Calculated quantities are written back into the HDF5 file.

In the third phase, the KITE-tools executable, a post-processing tool, reads the Chebyshev moments from the I/O file and uses them to compute the final quantities. The post-processing is the only part of the calculation that needs information about the free parameters in the formulae, such as the resolution $\eta$ in Green's functions and number of energy points. Consequently, it is possible to pass them as command-line arguments to the executable, ignoring those specific parameters in the configuration file, which can be useful, for example, for re-calculating the requested target functions for different temperatures or different chemical potentials.

It is also possible to use a smaller number of polynomials than those originally requested to KITEx, for example, to study convergence (figure 1). Each of the quantities that KITE-tools calculates has its own set of parameters that can be modified through command-line arguments. This scheme allows the user to calculate several quantities with a single KITEx usage. The post-processing usually takes less than 1 min.

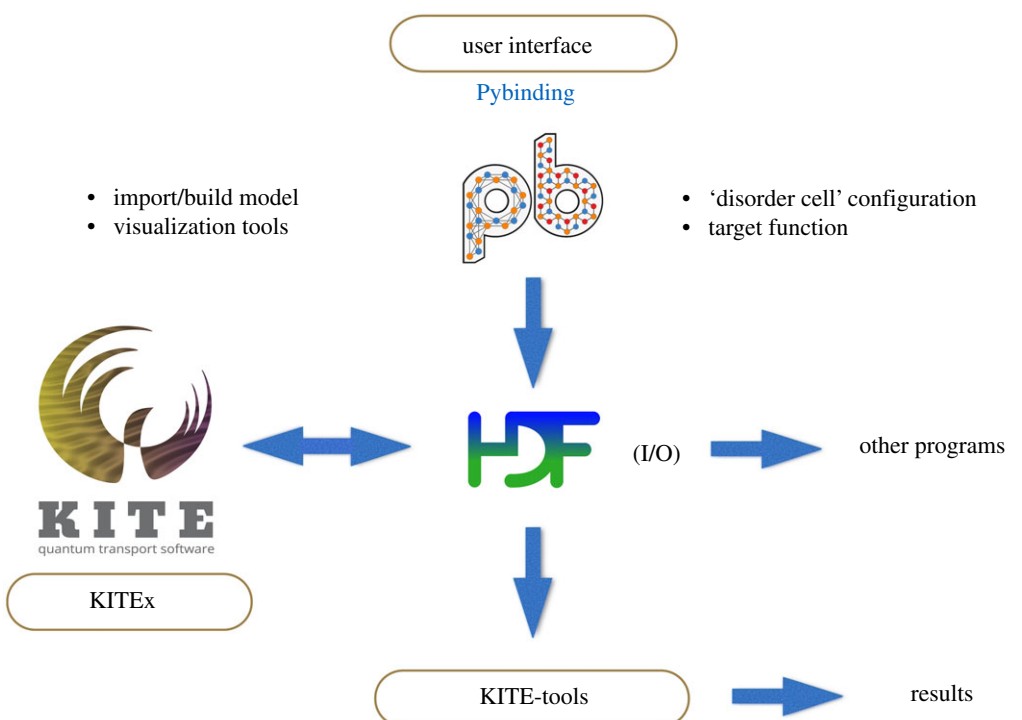

**Figure 3.** Code workflow. The user specifies the SETB model, disorder and other simulation parameters on a Python configuration script with Pybinding syntax. The script writes the information into a HDF5 file. The executable KITEx uses the HDF5 file as an input to perform the calculations. KITEx writes the resulting tables of Chebyshev moments to the original HDF5 file, which then serves as an input file for KITE-tools, a post-processing tool that provides the final data files for the calculated quantities. The HDF5 logo is adapted from [69].

## 3.4. Python interface

For setting up the tight-binding model, KITE relies on the Pybinding code developed by D. Moldovan at the University of Antwerp [47]. Pybinding provides a straightforward definition of orbitals and spin degrees of freedom, on-site energies and hopping parameters. A simple example script is shown below for the graphene honeycomb lattice with nearest neighbour hopping.

**Listing 1.** Setting up the tight-binding Hamiltonian.

```python
a = 0.24595  # [nm] unit cell length
a_cc = 0.142 # [nm] carbon-carbon distance
t = -2.8     # [eV] nearest neighbour pz-pz hopping

# create a lattice with 2 primitive vectors
lat = pb.Lattice(
    a1=[a, 0],
    a2=[a/2, a/2*sqrt(3)])

# add orbitals 'A' and 'B' (sublattices)
lat.add_sublattices(
    ('A', [0, -a_cc/2]),
    ('B', [0,  a_cc/2]))

# add hoppings
lat.add_hoppings(
    # inside the main cell
    ([0,  0], 'A', 'B', t),
    # between neighbouring cells
    ([1, -1], 'A', 'B', t),
    ([0, -1], 'A', 'B', t))
```

Next, the Python interface is used to set up the disorder configuration

**Listing 2.** Disorder configuration.

```
# define an object based on the lattice
disorder = kite.Disorder(lattice)
# add Gaussian distributed disorder at
# all sites of a given selected sublattice
disorder.add_disorder('A', 'Gaussian', 0.1, 0.1)
```

the system size and the decomposition domains,

**Listing 3.** System configuration.

```
# Number of domains, each of which is calculated in parallel
nx = ny = 4
# number of unit cells in each direction.
lx = ly = 512
# make config object which carries info about
# precision can be 0 - float, 1 - double, and 2 - long double.
configuration = kite.Configuration(divisions=[nx, ny], length=[lx, ly
    ], boundaries=[True, True], is_complex=False, precision=1)
```

and finally the type of calculation that will be performed by the C++ core. For example, for evaluating the $xx$ component of the optical conductivity tensor with $M = 512$ Chebyshev moments and $R = 20$ STE random vectors invoke the command

**Listing 4.** Simulation details.

```
calculation = kite.Calculation(configuration)
calculation.conductivity_optical(num_points=1000, num_disorder=1,
    num_random=20, num_moments=512, direction='xx')
```

Multiple calculations can be set up on the same script. Examples are covered in the next section. For more details about the usage of the post-processing tool and a complete list of functionalities and options, refer to the KITE website [20].

# 4. Examples

The following examples are also provided as script files in the examples folder from the Github repository of KITE [70].

## 4.1. Complex structures: low-angle twisted bilayer graphene

Twisted bilayer graphene (tBLG) systems with flat bands near the Fermi level provide a platform to explore strongly correlated phases and superconductivity [71–74]. Computational modelling of such systems is extremely demanding, especially at low twist angles [55]. As shown below, thanks to its ability to deal with very large systems, KITE enables studies of tBLG at 'magic angles' in real space with high probing resolution.

We focus on the largest magic angle, i.e. 1.05° [75]. We start from an SETB Hamiltonian for a non-interacting system

$$\hat{H} = -\sum_{i,j} t(\mathbf{r}_i, \mathbf{r}_j) \hat{c}_i^\dagger \hat{c}_j, \tag{4.1}$$

where $\hat{c}_i^\dagger$ ($\hat{c}_j$) is the electron creation (annihilation) operator on site $i(j)$. The transfer integral between the sites is taken as in [75,76]

$$-t(\mathbf{r}_i, \mathbf{r}_j) = V_{pp\pi} \left[ 1 - \left( \frac{\mathbf{d}_{ij} \cdot \mathbf{e}_z}{d} \right)^2 \right] + V_{pp\sigma} \left( \frac{\mathbf{d}_{ij} \cdot \mathbf{e}_z}{d} \right)^2, \tag{4.2}$$

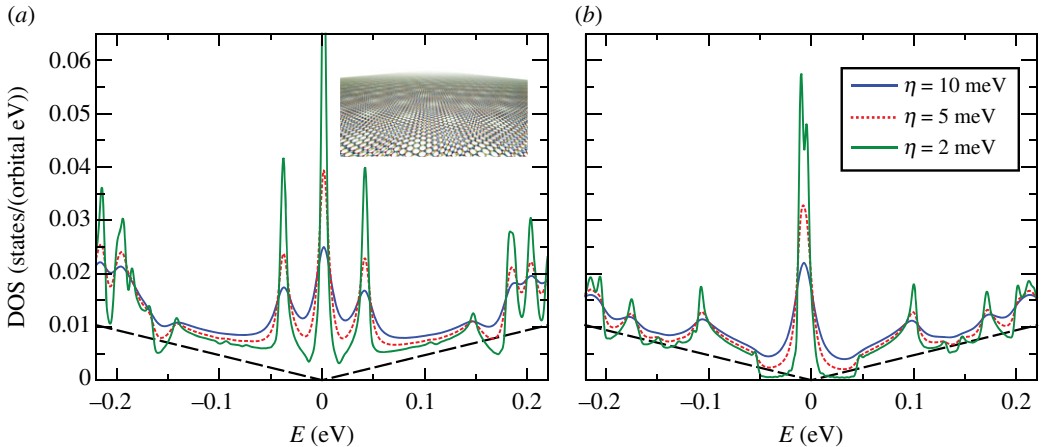

**Figure 4.** Comparison between the DOS of tBLG for (a) rigid non-relaxed lattice and (b) lattice relaxed by molecular dynamics, for different values of $\eta$. The inset of panel (a) depicts the tBLG lattice structure. Simulation details: $640 \times 512$ units cells, with 11 908 atomic sites per unit cell, $M = 12\,000$ and $R = 1$. Total RAM required $\approx 83$ GB (single real precision).

and
$$\begin{aligned} V_{pp\pi} &= V^0_{pp\pi} e^{-(d-a_0)/\delta} \\ V_{pp\sigma} &= V^0_{pp\sigma} e^{-(d-d_0)/\delta}, \end{aligned} \tag{4.3}$$

where $V^0_{pp\pi} = -2.7$ eV and $V^0_{pp\sigma} = 0.48$ eV are intralayer and interlayer hopping integrals, $a_0 \approx 0.142$ nm and $d_0 \approx 0.335$ nm are carbon–carbon distance in graphene and interlayer distance in bilayer graphene, respectively. $d_{ij}$ is the vector connecting two sites, $d = |d_{ij}|$ is the distance between them, and $\delta = 0.3187a_0$ is chosen in order to fit the next-nearest intra-layer hopping to $0.1V^0_{pp\pi}$. All neighbours within the distance of $4a_0$ are being considered.

The numerical analysis of special features in the electronic structure of tBLG systems, such as gap-opening and flat bands, require a small probing energy window to be resolved, usually of the order of 1 meV. To avoid finite-size effects, this resolution has to be finer than the mean-level spacing which originates from the discreteness of the simulated finite lattice. This brings, besides the intrinsic large unit cell of the twisted structures, the requirement for a large system. Due to the implementation of 'memory saving' algorithms explained in §3.2, KITE can handle such requirements. In this example, the simulated system contains $640 \times 512$ unit cells, with 11 908 atomic sites within one unit cell, which leads to a total number of orbitals $N \simeq 4 \times 10^9$, the largest SETB simulation reported in the literature to our knowledge. In addition to its giant dimension, the SETB model also has a very high coordination number $Z$ (around 60 neighbours).

Given the large size of the system, the STE can be safely undertaken using a single random vector for resolutions down to 1 meV. The DOS calculation can be requested with the following command

**Listing 5.** Code used for the DOS calculation.

```
calculation.dos(num_points=10000, num_moments=12000, num_random=1,
    num_disorder=1)
```

where a large number of moments ($M = 12\,000$) are requested to allow post-processing of DOS data with fine energy resolution (cf. figure 1d). KITEx code returns the set of calculated CPGF moments and KITE-tools reconstructs the DOS with the requested resolution.

Figure 4 shows the calculated DOS in two cases: a rigid, non-relaxed structure (a) and a lattice on which a molecular dynamics relaxation was performed (b). The relaxation was performed externally within the LAMMPS software package [77,78] by considering a combination of Brenner potentials for in-plane interactions and a registry-dependent Kolmogorov–Crespi potential [79] with parameters given in [80] for the out-of-plane van der Waals interaction. As suggested previously, refined features are resolved only at high-energy resolution ($\approx 1$ meV). Notably, the DOS peak at $E = 0$ eV indicates the presence of a flat band due to the specific twisting angle. Away from the flat band, the situation in the two cases is quite different. Mini-gaps around the flat bands start appearing only after relaxing the sample [81,82] and agree with recent experiments [83,84]. The $P$-complexity factor for this calculation (equation (2.35)) is $P = O(10^{15})$. Nevertheless, this calculation requires a modest 83 GB

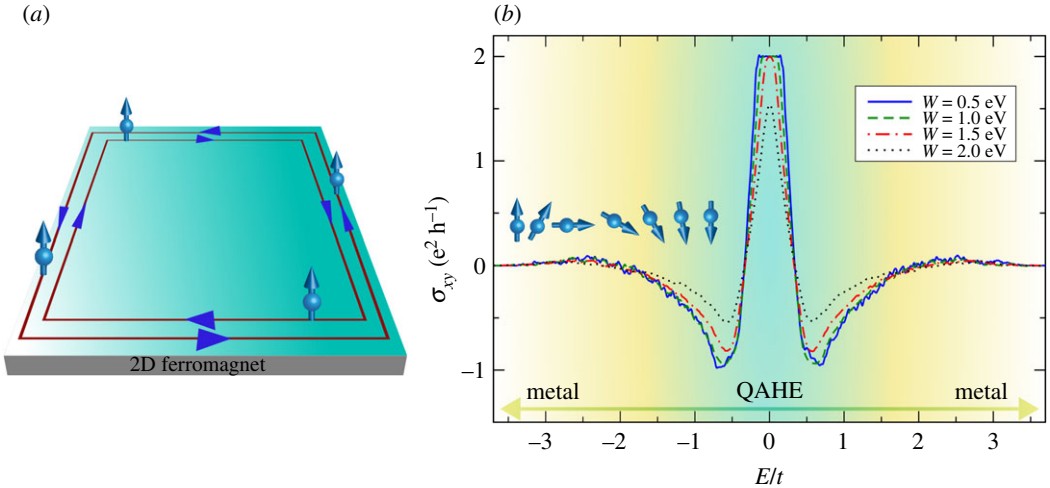

**Figure 5.** (a) Spin-polarized edge states of a two-dimensional quantum anomalous Hall insulator, (b) calculated Fermi energy dependence of the transverse charge conductivity for selected values of the disorder potential. The topological gap closes for $W \approx 2.0$ eV indicating a quantum phase transition driven by disorder fluctuations. Parameters: $\lambda_R = 0.3\,t$ and $\Delta_{ex} = 0.4\,t$. Simulation details: $8192 \times 8192$ unit cells and a total number of Chebyshev moments $M \times M$ with $M = 2048$. Results obtained for single random vector and disorder realization ($R$, $S = 1$). Minimum RAM required $\approx 8$ GB (double complex precision).

RAM using single real precision vectors for the STE evaluation, which means that even larger systems can be simulated if the calculations are run on special large memory nodes.

## 4.2. Topological materials

The scalability, accuracy and speed of KITE make it an ideal tool to simulate spectral properties and response functions of materials with non-trivial band topology [53,67]. To illustrate this capability, we consider the quantum anomalous Hall insulating regime of a magnetized graphene monolayer with interfacial broken inversion symmetry. The minimal model, incorporating SOC, proximity-induced magnetic exchange and scalar disorder [85,86], is given by

$$\hat{H} = -t \sum_{\langle ij \rangle, s} \hat{c}_{is}^\dagger \hat{c}_{js} + \frac{2i}{3} \sum_{<i,j>,s,s'} \hat{c}_{is}^\dagger \hat{c}_{js'} [\lambda_R (\hat{\mathbf{s}} \times \mathbf{d}_{ij})_z]_{ss'} + \Delta_{ex} \sum_{i,s} \hat{c}_{is}^\dagger \hat{s}_z \hat{c}_{is} + V_{dis}. \tag{4.4}$$

The first term describes nearest-neighbour hopping processes ($\hat{c}_{is}^\dagger$ adds electrons with spin state $s = \uparrow, \downarrow$ to site $i$). The second term is the Bychkov–Rashba interaction with coupling strength $\lambda_R$. $\mathbf{d}_{ij}$ is the unit vector pointing from site $j$ to $i$ and $\hat{\mathbf{s}}$ is the vector of Pauli matrices. The third term describes a uniform exchange field with strength $\Delta_{ex}$. The last term stands for the disorder potential (see below). The interplay of Bychkov–Rashba spin-orbit coupling (BRSOC) and exchange field endows the electronic states of the clean system with non-coplanar spin texture in momentum space [86] and opens a topologically non-trivial insulating bulk gap [85]. The exchange field breaks time-reversal symmetry and the corresponding insulating phase is characterized by a Chern number $\mathcal{C} = 2$ in the bulk, with spin-polarized states protected against elastic backscattering at the edges (figure 5a). Consequently, the quantum anomalous Hall insulator of the clean system has quantized Hall conductivity of $\sigma_{xy} = 2e^2/h$ inside the gap.

Several types of disorder can be defined in the configuration file. Given its high efficiency, KITE allows to assess the robustness of the topological phase directly from the behaviour of the Hall conductivity at a modest computational cost. For illustration purposes, we model $V_{dis} = \sum_{i,s} V_{i,s} \hat{c}_{is}^\dagger \hat{c}_{is}$ as a white-noise random potential distributed on the box $V_{i,s} \in [-W/2, W/2]$. To make the disorder spin-independent and thus time-reversal symmetric, $V_{i,\uparrow} = V_{i,\downarrow}$, one separates $A$ and $B$ sublattices on kite.configuration, using

**Listing 6.** Specifying the spin-independent random disorder potential.

```
disorder = kite.Disorder(lattice)
disorder.add_disorder(['Aup', 'Adown'], 'Uniform', 0.0, W)
disorder.add_disorder(['Bup', 'Bdown'], 'Uniform', 0.0, W)
```

Finally, one needs to set kite.calculation. The post-processing tool requires spectrum endpoints to be specified $E \in [E_{\min}, E_{\max}]$, so that Fermi sea integrations can be carried out avoiding spurious effects. Tight spectral bounds are generally hard to extract, so it is recommended to couple the Hall conductivity with average DOS calculations using

**Listing 7.** Code used for the DOS and DC conductivity calculations.

```
calculation.dos(num_points=6000, num_moments=2048, num_random=1,
    num_disorder=1)
calculation.conductivity_dc(num_points=6000, num_moments=2048,
    num_random=1, num_disorder=1, direction='xy', temperature=1)
```

KITE estimates the spectrum endpoints $E_{\max(\min)}$ by using a small system built from the defined Hamiltonian, unless pre-defined values are specified (see equations (2.14) and (2.15)). This automated estimation procedure *does not* necessarily guarantee that the transformed spectrum of the large-scale system will fall inside the canonical interval $\epsilon \in [-1 : 1]$. We recommend that users inspect DOS curves obtained with the automatic rescaling in order to validate the estimated endpoints.

The Hall conductivity functionality implements a full-spectral expansion of the Kubo–Bastin formula, where KITEx computes $M \times M$ Chebyshev moments, and KITE-tools uses the $\Gamma$ matrix to reconstruct $\sigma_{xy}$ over any desired energy integral. The energy resolution is limited by the computational domain size and number of moments retained (§2). Both temperature and num_points are parameters used by KITE-tools, and it is possible to modify them without re-running KITEx. This type of calculation requires more memory than a DOS or single-shot DC conductivity computation. Nevertheless, an efficient memory management enables to reach large system sizes. Figure 5b shows the $T = 0$ transverse conductivity $\sigma_{xy}$ for a lattice of $8192 \times 8192$ unit cells and a total number of Chebyshev moments $2048 \times 2048$. The corresponding $P$-complexity factor is $P = O(10^{15})$. KITE captures the anomalous quantum Hall plateau extremely well, with a relative error of less than 1%. The behaviour in the metallic regime follows the many-body theoretical predictions for magnetized two-dimensional materials with dilute disorder [86]. Moreover, the CPGF approach is not limited to the diffusive transport regime of weak disorder, so it describes accurately the closing of the topological gap with increasing disorder strength. The results in figure 5b indicate a critical disorder strength of about $W \approx 2.0$ eV for the simulated system.

## 4.3. Linear and nonlinear optical response

One of the highlights of the open-source KITE package is its efficient numerical implementation of optical response functions $\hat{\sigma}(\omega)$. It provides the framework for tackling generic multi-orbital SETB models in the presence of disorder, defects, strain and even external magnetic fields. KITEx spectral algorithms give access to unprecedented large system sizes with accessible energy resolutions only limited by the mean level spacing. This opens up the possibility to tackle problems previously considered extremely challenging or even unfeasible, such as capturing mgneto-transport effects in two-dimensional lattices threaded by small magnetic fluxes; see §4.6. To illustrate some of these new capabilities, we invoke the optical conductivity target function

**Listing 8.** Code used for the optical conductivity calculation.

```
calculation.conductivity_optical(num_points=6000, num_moments=2048,
    num_random=1, num_disorder=1, direction='xx')
calculation.conductivity_optical(num_points=6000, num_moments=2048,
    num_random=1, num_disorder=1, direction='xy')
```

The time-consuming recursive calculation is independent on photon frequency and Fermi energy (see equation (2.26)). Similar to the average DOS functionality (§2), KITE-tools can modify both parameters *a posteriori*, without re-calculating Chebyshev matrices $\Gamma_{nm}$. Figure 6 shows $\sigma_{xx}(\omega)$ and $\sigma_{xy}(\omega)$ calculated for the Hamiltonian in equation (4.4). Many thousands of Chebyshev moments are required to properly converge the optical response at low frequencies (DC regime).

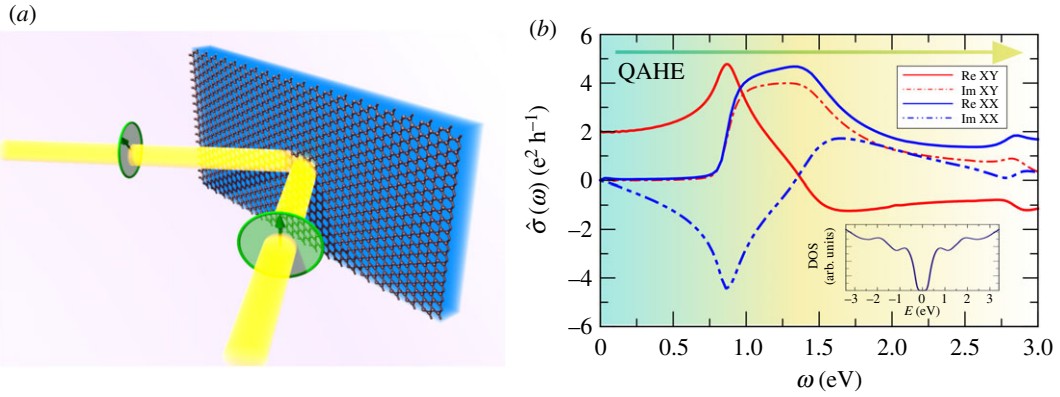

**Figure 6.** (a) Linearly polarized beam reflected from a magnetic active material and respective Kerr effect. (b) $\sigma_{xx}(\omega)$ and $\sigma_{xy}(\omega)$ of a quantum anomalous Hall insulator with Fermi energy inside the gap ($E = 0$ eV). The inset shows the DOS in a $\pm 3$ eV window around the Dirac ($K$) point. Simulation details: $8192 \times 8192$ unit cells and a total number of Chebyshev moments $M \times M$ with $M = 6000$. Model parameters: $W = 1.25$ eV; other parameters as in figure 5. Minimum RAM required $\approx 8$ GB (double complex precision).

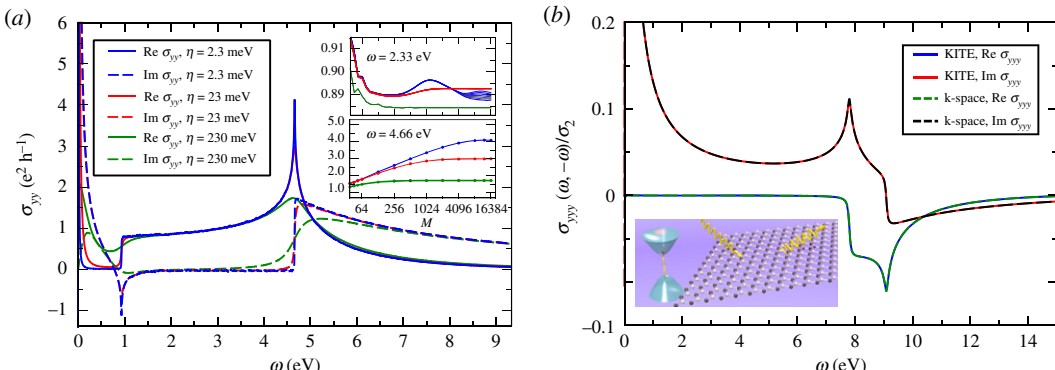

**Figure 7.** (a) First-order optical conductivity of clean graphene as a function of photon energy at selected values of energy resolution $\eta$. The solid (dashed) curves represent the real (imaginary) part of the conductivity at $T = 0$. The lower (upper) inset shows the evolution of the conductivity as the number of polynomials is increased for multiple closely spaced photon energies around $\hbar\omega = 2.33$ (4.66) eV. The lack of convergence for $\eta = 2.3$ meV (upper panel) indicates that the discrete nature of the spectrum is being resolved. Simulation details: $2048 \times 2048$ unit cells, total number of Chebyshev moments $M \times M$ with $M = 16384$ and one random vector $R = 1$. Hopping parameter: $t = 2.33$ eV and Fermi energy $E = 0.466$eV (figure adapted from João & Lopes [58]). Total RAM required $\approx 2.25$ GB (double complex precision). (b) Second-order $yyy$ photogalvanic effect for clean h-BN. Model parameters: $t = 2.33$ eV, $E = 0$ eV, energy gap $\Delta = 7.80$ eV, and $\eta = 39$ meV. Here, $\sigma_2 \equiv e^3 a/4th$. Simulation details: total number of Chebyshev moments $M^3$ with $M = 2048$, $4096 \times 4096$ unit cells, and $R = 1$. Total RAM required $\approx 64$ GB (double precision). Inset shows an illustration of the electron–photon interaction.

From high-resolution traces of $\hat{\sigma}(\omega)$, it is possible to extract interesting properties, such as the Faraday rotation angle [87–89], experimentally accessible in magneto-optical measurements [90]. The post-processing with KITE-tools can be performed with a subset of the calculated Chebyshev moments. With this feature, users can analyse the spectral convergence with a single KITEx simulation. Similarly, it is possible to modify $\eta$ with KITE-tools and quickly re-calculate the target functions for different energy resolutions.

Figure 7a shows the $yy$ optical conductivity for a clean monolayer graphene sample with mean level spacing $\delta\varepsilon = 5.3$ meV at the Dirac ($K$) point at selected values of the energy resolution parameter $\eta$. The lower inset shows the convergence as a function of the number of polynomials for $\hbar\omega = 4.66$ eV, a region of rapidly changing conductivity. Clearly, calculations with higher energy resolution require substantially more polynomials to converge (see equation (2.20). For meaningful resolutions $\eta \gtrsim \delta\varepsilon$, the optical conductivity curves with $M \approx 10\,000$ are converged to accuracy of 1% or better. For pedagogical reasons, we show a calculation with $\eta = 2.3$ meV $< \delta\varepsilon$. In this case, the discreteness of the spectrum becomes noticeable through spikes in the conductivity.

The upper inset shows a similar analysis, but now for a tiny region around $\hbar\omega = 2.33$ eV, a region of slowly increasing conductivity. The plot shows three sets of curves with different colours. Inside each set, we represent a collection of frequencies, ranging from $\hbar\omega = 2.3300$ eV to $\hbar\omega = 2.3316$ eV. The darker curves correspond to higher frequencies. The main graph shows that all of these curves have converged to the same value in a region of slowly increasing conductivity. The inset, however, shows a different picture. The red ($\eta = 23$ meV) and green ($\eta = 230$ meV) sets of curves show a variation consistent with the expected increase of the conductivity. If one zooms in to those sets of curves, it is possible to notice that they are indeed increasing in value with $\omega$. The blue curve ($\eta = 2.3$ meV) is not only changing on a scale much more significant than expected, but it is also decreasing. This effect, shown here for pedagogical purposes, comes from the artificial choice of $\eta$ (which is incompatible with the small size of the simulated system).

### 4.3.1. Nonlinear response

The Kubo formula can be expanded to arbitrary order using the Keldysh formalism (equation (2.21)). To exemplify the evaluation of the nonlinear optical conductivity, we consider the second-order conductivity $\sigma^{yyy}(\omega, -\omega)$ of hexagonal boron nitride (h-BN) [58]. To calculate the quantity, we invoke the following command:

**Listing 9.** Code used for the calculation of the second-order conductivity.

```
calculation.conductivity_optical_nonlinear(num_points=2000,
    num_moments=2048, num_random=1, num_disorder=1, direction='yyy')
```

The target function depends on two frequencies $\omega_1$ and $\omega_2$, which need only to be specified at the post-processing level. The numerical results for the clean system are shown in figure 7b, and compared with numerically exact results obtained from $k$-space integration [91]. The physically meaningful symmetrized response $(\sigma^{yyy}(\omega, -\omega) + \sigma^{yyy}(-\omega, \omega))/2$ is purely real; see equation (2.24). The photogalvanic nonlinear conductivity is non-zero when the photon energy exceeds the gap ($\hbar\omega \gtrsim \Delta = 7.80$ eV) and peaks around 9 eV [91].

## 4.4. Electronic structure: spectral functions and spatial LDOS

KITE can compute local density of states (LDOS) and spectral functions of direct relevance for angle-resolved photoemission spectroscopy (ARPES). As shown in what follows, the calculations of real-space and momentum-space spectral quantities rely on a single Green's function Chebyshev expansion and, as such, are fast and accurate.

### 4.4.1. Spectral functions

For the calculation of spectral function, instead of using a random vector to compute the density of states, KITE uses a specific vector $k$, defined in the Brillouin zone

$$\rho_k(\epsilon) = \frac{1}{N} \langle k | \delta(\epsilon - \hat{H}) | k \rangle, \tag{4.5}$$

where $|k\rangle$ is a sum of Bloch vectors

$$|k, \alpha\rangle = \sum_i \exp(i k \cdot R_i^\alpha) |i, \alpha\rangle, \tag{4.6}$$

weighted by a structure factor $w_\alpha(k)$

$$|k\rangle = \sum_\alpha w_\alpha(k) |k, \alpha\rangle, \tag{4.7}$$

where $i$ runs through all lattice sites and $\alpha$ labels the orbitals. The structure factor is formally given by the Fourier transform of the localized wavefunctions $w_\alpha(\mathbf{r})$ [92]. The intensity of the response depends on the specific form of $w_\alpha(k)$ and cannot be accurately described without it. However, if the system possesses

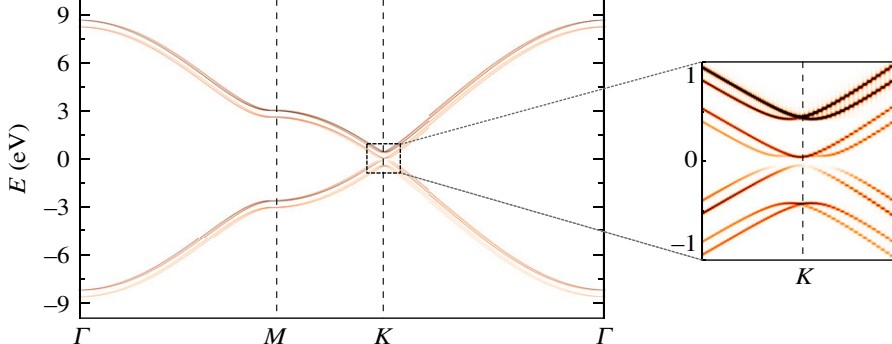

**Figure 8.** Spectral function of an electrically biased AB graphene bilayer with Bychkov–Rashba interaction in the presence of Anderson disorder ($W = 0.07t$) based on a modified version of the spectral function module with a single value of orbital weights, which reflects in the non-equal contribution of different bands. The spin splitting of bands caused by interfacial broken inversion symmetry and broadening of spectral lines due to disorder are clearly visible. Model parameters [93]: $t = -2.8$ eV, $t_\perp = -0.4$ eV, $U = 0.05$ eV and $t_R = \lambda_R$ with $\lambda_R = 0.075$ eV. Simulation details: $1024 \times 1024$ unit cells with $M = 4096$ (corresponding to an energy resolution of $\approx 10$ meV). Total RAM required $\approx 200$ MB (double complex precision).

translation symmetry, its band structure may be computed by averaging over a distribution of $w_\alpha(\mathbf{k})$. For this, one needs to express the $|\mathbf{k}, \alpha\rangle$ states in terms of the band states

$$|\mathbf{k}, \alpha\rangle = \sum_n |\mathbf{k}, n\rangle\langle \mathbf{k}, n|\mathbf{k}, \alpha\rangle. \tag{4.8}$$

Then, $\rho_{\mathbf{k}}(\varepsilon)$ takes the form

$$\rho_{\mathbf{k}}(\epsilon) = \sum_{\alpha\beta n} w_\alpha^*(\mathbf{k})w_\beta(\mathbf{k})\langle \mathbf{k}, n|\mathbf{k}, \beta\rangle\langle \mathbf{k}, \alpha|\mathbf{k}, n\rangle\delta(\epsilon - \epsilon_n(\mathbf{k})). \tag{4.9}$$

Averaging over the orbital weights with $\langle w_\alpha^*(\mathbf{k})w_\beta(\mathbf{k})\rangle = \delta_{\alpha,\beta}$, we obtain

$$\langle \rho_{\mathbf{k}}(\epsilon)\rangle_w = \frac{1}{N}\sum_n \delta(\epsilon - \epsilon_n(\mathbf{k})). \tag{4.10}$$

All the bands have the same intensity which is independent of the $\mathbf{k}$ point. If the system does not possess translation symmetry, then $\langle \rho_{\mathbf{k}}(\epsilon)\rangle_w$ will no longer be related to the band structure.

As an example, we consider a Bernal (AB)-stacked graphene bilayer subject to a perpendicular (bias) electric field [38,93,94]. We use KITE's in-built spectral function capability to reverse engineer the band structure and resolve the disorder-induced broadening of quasi-particle states. In the absence of a bias, the system is a semimetal, and the band structure presents four spin-degenerate bands (in the absence of BRSOC). Figure 8 shows that application of a small bias opens up a gap in the spectrum, while a finite BRSOC lifts the spin degeneracy and endows the energy bands with a spin-helical structure [86].

**Listing 10.** Code used for the spectral function calculation.

```
b1, b2 = lattice.reciprocal_vectors()
Gamma = [0, 0]
K = 1 / 3 * (b1[0:2] + b2[0:2])
M = b1[0:2]/2
points = [Gamma, M, K, Gamma]
k_path = pb.results.make_path(*points[path_num], step=0.005)
weights = [10.0, 5.0, 78.0, 16.0, 1.0, 0.01, 31.0, 9.0]
configuration = kite.Configuration(divisions=[4, 4], length=[1024,
    1024], boundaries=[True, True], is_complex=True, precision=1,
    spectrum_range=[-10,10])
calculation_arpes = kite.Calculation(configuration)
calculation_arpes.arpes(k_vector=k_path, num_moments=4096, weight=
    weights, num_disorder=1)
```

## 4.4.2. Local density of states

In a similar manner, the local density of states is calculated by finding the expectation value of projection of the spectral operator onto a specific orbital

$$\rho_{i\alpha}(\epsilon) = \langle i, \alpha | \delta(\epsilon - \hat{H}) | i, \alpha \rangle. \tag{4.11}$$

To illustrate the calculation of orbital-projected local density of states with KITE, we consider the effect of a single vacancy in a monolayer of $WS_2$. To better understand the orbital projection, it is illustrative to use a six-orbitals model containing three orbitals for the transition metal and three orbitals for the chalcogen atom, as presented in [95]. We consider the effect of a transition metal (TM) vacancy at a given site $i$. We calculate the LDOS on the TM sublattice and resolve its three different contributions ($d_{3z^2-r^2}$, $d_{xy}$ and $d_{x^2-y^2}$).

**Listing 11.** Code used for the LDOS calculation.

```python
#Location of defect [[x=d1,y=d2]]
d1 = d2 = 32
#pos_matrix and sub_matrix: list of positions for LDOS calculation
pos_matrix=[]
sub_matrix=[]
lim = 10
for i in range(-lim,lim):
  for j in range(-lim,lim):
    pos_matrix.append([d1+i,d2+j])
    pos_matrix.append([d1+i,d2+j])
    pos_matrix.append([d1+i,d2+j])
    sub_matrix.append('M1')
    sub_matrix.append('M2')
    sub_matrix.append('M3')
#Define the structural disorder (TM vacancy)
struc_disorder_1 = kite.StructuralDisorder(lattice, position=[[d1,d2
    ]])
struc_disorder_2 = kite.StructuralDisorder(lattice, position=[[d1,d2
    ]])
struc_disorder_3 = kite.StructuralDisorder(lattice, position=[[d1,d2
    ]])
struc_disorder_1.add_vacancy('M1')
struc_disorder_2.add_vacancy('M2')
struc_disorder_3.add_vacancy('M3')
disorder_structural = [struc_disorder_1, struc_disorder_2,
    struc_disorder_3]
#Domain decomposition and system size
nx = ny = 4
lx = ly = 2048
#System configuration simulation type (LDOS)
configuration = kite.Configuration(divisions=[nx, ny], length=[lx, ly
    ], boundaries=[True, True], is_complex=True, precision=1,
    spectrum_range = [-10,10])
calculation = kite.Calculation(configuration)
calculation.ldos(energy=np.linspace(-0.1, 0.1, 3), num_moments=1024,
    num_disorder=1, position=pos_matrix, sublattice=sub_matrix)
```

Results of the LDOS calculation are shown in figure 9.

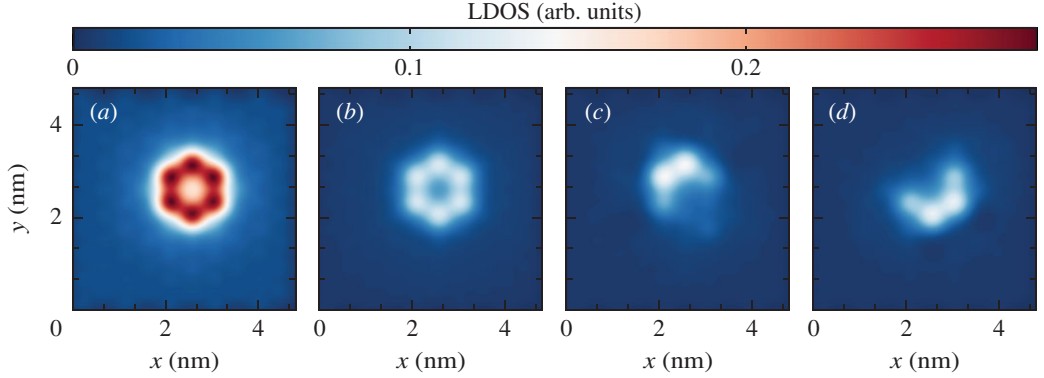

**Figure 9.** LDOS of a vacancy in WSe$_2$ for $E = 0$ eV, with a tight-binding model of six orbitals [95]: (a) the total LDOS, and the orbital projected LDOS for orbitals (b) d$_{3z^2-r^2}$, (c) d$_{xy}$ and (d) d$_{x^2-y^2}$. Simulation details: $256 \times 256$ unit cells and $M = 512$ Chebyshev moments. Total RAM required $\approx 10$ MB (double complex precision).

## 4.5. Spintronics: time-evolution of spin polarized wave-packets

KITE also provides the functionality of performing real-space wave-packet propagation [96]. The time evolution of wave-packets $|\psi(t)\rangle$ under a time-independent Hamiltonian is determined by the time-evolution operator $\hat{U}(t)$ according to

$$|\psi(t)\rangle = e^{-i\hat{H}t/\hbar}|\psi(0)\rangle = \hat{U}(t)|\psi(0)\rangle, \tag{4.12}$$

where $|\psi(0)\rangle$ is the initial state. The spectral expansion of the time-evolution operator in terms of Chebyshev polynomials of first kind is given by [97]

$$\hat{U}(t) = e^{-i\hat{H}t/\hbar} = e^{-i\delta\epsilon_+ t/\hbar}\left[c_0 + 2\sum_{n=1}^{\infty} c_n \mathcal{T}_n(\hat{H})\right], \tag{4.13}$$

where $\hat{H}$ is the rescaled Hamiltonian with eigenvalues in the canonical interval (see equations (2.14) and (2.15)) and $c_n = (-1)^n J_n(\delta\epsilon_- t/\hbar)$, where $J_n(\delta\epsilon_- t/\hbar)$ are Bessel functions of the first kind.

To exemplify this functionality, we use KITE to resolve the spin dynamics in heterostructures of graphene and semiconducting (group VI) transition metal dichalcogenide (TMD) monolayers. The point group symmetry is $C_{3v}$. Hence, two types of interface-induced spin–orbit effects are allowed [98,99]: (i) intrinsic-like SOC (invariant under the crystal symmetries of the isolated monolayers), and (ii) Bychkov–Rashba interaction due to broken mirror reflection ($z \rightarrow -z$) symmetry. The characteristic spin–valley coupling of the TMD monolayer [100] is 'imprinted' on graphene states, becoming the dominant interfacial intrinsic-type SOC [101]. This spin–valley coupling acts as a pseudomagnetic field oriented along $\hat{z}$ for electrons at $K(K^{'})$ Dirac points, leading to highly anisotropic spin dynamics characterized by long out-of-plane spin lifetimes [102–104]. Such a spin relaxation anisotropy was recently observed in Hanle-type spin precession measurements on graphene/TMD bilayers [105,106]. To simulate spin-relaxation dynamics in the presence of disorder, we consider a nearest-neighbour SETB model for monolayer graphene subject to Bychkov–Rashba effect

$$\hat{H} = -\sum_{<i,j>,s} t\,\hat{c}_{is}^\dagger \hat{c}_{js} + \frac{2i}{3}\sum_{<i,j>,s,s'} \hat{c}_{is}^\dagger \hat{c}_{js'}[\lambda_R(\hat{s} \times \mathbf{d}_{ij})_z]_{ss'} + \sum_{i,s} \Delta_{is}\,\hat{c}_{is}^\dagger \hat{c}_{is}, \tag{4.14}$$

where the first two terms are defined below equation (4.4) and the last term accounts for an on-site potential that can represent either an Anderson disorder or a magnetic impurity. In this example, we illustrate how the spin dynamics of in-plane and out-of-plane spins is affected by both types of disorder. Section 4.2 presented the on-site disorder entry in KITE, with $\Delta_{is} \in [-W/2, +W/2]$. Magnetic impurities are modelled as resonant Ising scatterers with opposite sign for each spin, $\Delta_{is} = W_{res}s_z$ (if $i$ is occupied by an impurity), distributed with a given concentration $c$. The latter can be incorporated with the following command:

**Listing 12.** Code used for implementing resonant magnetic impurities.

```
# Aup(down) and Bup(down) are spin-orbitals
conc = 0.04 #concentration of impurities
struc_disorder_one = kite.StructuralDisorder(lattice,
concentration=conc)
struc_disorder_two = kite.StructuralDisorder(lattice,
concentration=conc)
struc_disorder_one.add_structural_disorder(
# in this way we can add onsite disorder in the form
#"[unit cell], 'sublattice', value"
    ([0, 0], 'Aup'   , +w_res),
    ([0, 0], 'Adown', -w_res),
)
# One can add multiple disorder types, which should be forwarded
to the export_lattice function as a list.
struc_disorder_two.add_structural_disorder(
    # in this way we can add onsite disorder in the form [unit
cell], 'sublattice', value
    ([0, 0], 'Bup'   , +w_res),
    ([0, 0], 'Bdown', -w_res),
)
```

An initial spin-polarized wave-packet can be constructed in the following manner:

$$|\Psi(0, \mathbf{r})\rangle = e^{-1/2\mathbf{r}^2\sigma^2} \sum_{\mathbf{k}_j} e^{i\mathbf{k}_j\cdot\mathbf{r}}|\psi(\mathbf{k}_j)\rangle_S, \tag{4.15}$$

where $|\psi(\mathbf{k}_j)\rangle_S$ is a spinor defined over the full Hilbert space, and which has a well-defined spin expectation value. Since in graphene the pseudo-spin, manifest in the sublattice polarization, is aligned with the momentum, this spinor will depend on $\mathbf{k}_j$.

For the use of the time-evolution module, it is necessary to define the lattice and type of disorder, as usual, and invoke the time-evolution functionality:

**Listing 13.** Code used for the wave-packet time evolution calculation.

```
calculation.Gaussian_wave_packet(num_moments=num_moments,
    num_disorder=1, k_vector=k_vector, spinor=spinor, width=sigma,
    num_points=num_points, mean_value=[int(lx / 2), int(ly / 2)],
    timestep=scale_a * deltaT / hbar)
```

where the parameters k_vector and spinor are lists of sampling vectors, $\mathbf{k}_j$, in reciprocal space and the corresponding spinors. Width is the parameter which defines the standard deviation in real space of the 'Gaussian' wave-packet at $t = 0$, and mean_value specifies around which point (in unit cell coordinates) the wave-packet is centred. Additional parameters for this type of calculation are num_points and timestep, which represent the number of timepoints at which the observables are computed, and the time step $\Delta t$, respectively.

Within KITEx, the following observables are calculated:

— the spin expectation along $x$-, $y$- and $z$-directions, $S_{x,y,z} = \langle\Psi(t)|\hat{\sigma}_{x,y,z}|\Psi(t)\rangle$, where $\hat{\sigma}_{x,y,z}$ are the Pauli matrices, returned in the units of $\hbar/2$,
— the mean displacement, or the mean position of the wave-packet in all spatial directions, $\langle\Psi(t)|\hat{x}, \hat{y}, \hat{z}|\Psi(t)\rangle$, returned in the units of the specified lattice vectors, usually nm,
— the mean square displacement, $\langle\Psi(t)|\hat{x}^2, \hat{y}^2, \hat{z}^2|\Psi(t)\rangle$, returned in the square of the units of the specified lattice vectors, usually nm$^2$.

The output of the wave-packet simulation (i.e. mean displacement, mean square displacement and spin expectation values) is directly written to the HDF5 file.

For a typical spin dynamics simulation, the energy uncertainty of the wave-packet $\sigma_E$ needs to be chosen much smaller than significant energy scales in the model. This guarantees that the computed

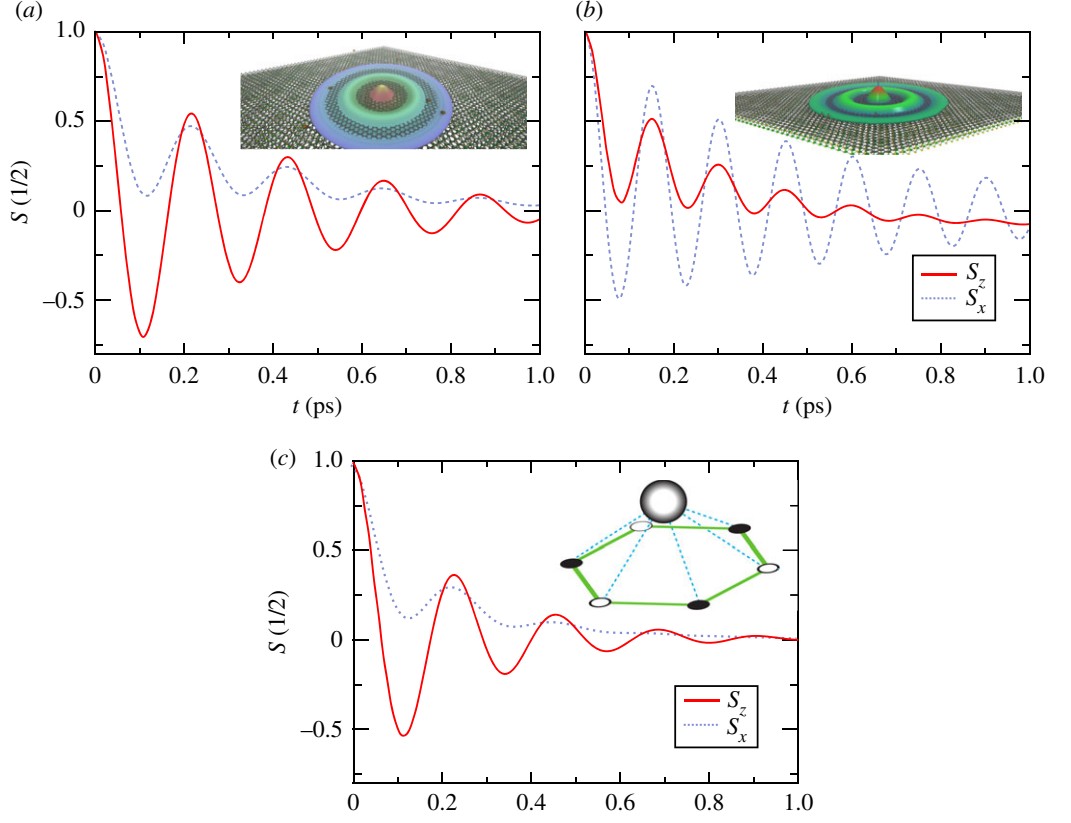

**Figure 10.** Time evolution of in-plane ($S_\parallel$) and out-of-plane ($S_\perp$) spin components of a spin-polarized wave-packet in a medium-size graphene/TMD heterostructure flake with $\approx 103$ million ($p_z$-) orbitals in the presence of (*a*) Anderson disorder with $W = 1.5$ eV and Anderson disorder ($W = 1.5$ eV) superimposed with (*b*) magnetic short-range impurities ($W_{\text{res}} = 0.25$ eV) (non-magnetic heavy adatoms in hollow position (*c*)) with concentration $c = 0.04$. The average self-energy mediated by the impurities $\Delta_{\text{ex}} = c^* W_{\text{res}}$ matches the SOC energy scale, thus strongly affecting the spin dynamics. In (*c*), the self-energy mediated by the heavy adatoms give rise to several spin–orbit interactions within the adsorption sites. The adatom–graphene interaction is parametrized by a set of spin-conserving hoppings (nearest neighbour, $t_{\text{imp}} = -0.85 \mp 0.08\,\text{i}$ eV, second-nearest neighbour, $\nu_{\text{imp}} = -0.49 \mp 0.08\,\text{i}$ eV, and third-nearest neighbour, $\rho_{\text{imp}} = -0.35$ eV, where $\pm$ holds for spin up (down) electrons), as well as spin–flip interactions between all sites of the hexagonal plaquette ($\lambda_{\text{imp}} = -0.12$ eV) due to the broken mirror reflection symmetry (for details, see [66]). Other parameters read as: $t = -2.507$ eV, $\lambda_R = 10$ meV, Fermi energy $E = 0.1$ eV and initial wave-packet width $\sigma \approx 110$ nm. The insets illustrate snapshots of the real-space wave-packet profile (*a,b*) and a hollow adatom (*c*). Simulation details: $7168 \times 7168$ unit cells, time steps of 2 fs and $M = 70$ Chebyshev moments per time step. Total RAM required $\approx 17$ GB (double complex precision).

spin dynamics is limited by elastic scattering processes at the Fermi surface leading to spin relaxation (and not by energy dephasing). We note that the constraint $\sigma_E \ll \lambda_R$ leads to the requirement of a significant standard deviation $\sigma$, and in turn a large computational domain (system size).

To illustrate this capability, we simulate the spin dynamics of graphene on a TMD monolayer in the strong SOC regime ($\lambda_R \tau \gg \hbar$, where $\tau$ is the scattering time [104]), subject to different types of disorder. The BRSOC acts as an in-plane pseudo-magnetic field, and both $S_\parallel \equiv S_{x,y}$ and $S_\perp \equiv S_z$ are subjected to precession. In our computational experiment, the originally defined wave-packet at $t = 0$ (i.e. a sum of plane waves with a Gaussian envelope) diffuses isotropically in the basal plane of graphene in the presence of Anderson scalar disorder only (figure 10*a*) and Anderson disorder combined with a resonant concentration of magnetic impurities (figure 10*b*) or hollow adatoms (figure 10*c*). In all cases, the spin precession is anisotropic and the spin density executes several precession cycles around the pseudo-magnetic field before fading way. In this strong SOC regime, the spin dynamics is characterized by fast damped oscillations, with spins relaxing on the timescale of a single scattering event. For example, for short-range scalar disorder (*a*), the two spin components are precessing accordingly with predictions, and KITE captures the $\cos(\lambda_R \tau/\hbar)$ evolution of the $S_\perp$ spin, as well as the fine effect of the higher-order precession terms which result in the $\cos^2(\lambda_R \tau/\hbar)$ evolution of $S_\parallel$

[104]. In the presence of magnetic (Ising) impurities (*b*), the spin precesses about two different axes, one coming from the in-plane uniform BRSOC, and another with origin in the out-of-plane field induced by the impurities, which explains the qualitative change in the time evolution of the spin expectation values. On the other hand, panel (*c*) shows an example where the spin dynamics is influenced by non-magnetic heavy adatoms (e.g. thallium) leading to several competing short-range interactions [66]. The complex adatom-graphene Hamiltonian, comprising two spin-conserving SOC hoppings, a local BRSOC and third-nearest neighbour hopping, modifies the spin dynamics (most noticeably by reducing the lifetime of in-plane spins) with respect to samples (*a*,*b*).

## 4.6. Magnetic field

KITE allows to compute response functions in the presence of an external magnetic field with the use of Peierls substitution method [107], which depends on the lattice structure and hopping matrices. In the current version, this functionality is only available for lattices with periodic boundary conditions. Hence, the achievable magnetic flux per unit cell is limited by the system size [107]. However, the simplicity and flexibility of KITE in implementing SETB models is extended to the Peierls substitution that is *automatically defined* for any lattice structure in two dimensions. To invoke the automated magnetic field functionality, one needs to include a modification function in the configuration file, as shown below

**Listing 14.** Specifying the magnetic field.

```
mod = kite.Modification(magnetic_field = 10)
kite.config_system(lattice, configuration, calculation, modification=
    mod, filename='magnetic.h5')
```

where it is possible to define either the magnetic field (in tesla) or the magnetic flux per unit cell (in unit of flux quantum, $h/e$). When KITE generates the HDF5 file, it automatically calculates the magnetic field that best matches the one defined by the user. In previous studies, real-space (tight-binding) approaches were limited to small system sizes, and the Peierls substitution in periodic systems gives rise to unrealistic large magnetic fields. The efficient spectral algorithms implemented in KITE enable high-resolution spectral calculations in extremely large lattices. Consequently, one can perform calculations with realistic magnetic fields using modest computational resources. We exemplify this functionality by considering a single layer of black phosphorus [108]. Although the SETB model implemented is relatively simple, it contains four orbitals and five hopping terms in a unit cell and illustrates well the generality of our Peierls substitution. In the presence of a perpendicular magnetic field, the energy spectrum consists of equally spaced Landau levels in both electron and hole sectors. However, we note that the energy level separation is different for electrons and holes due to their different anisotropic effective masses [108,109].

Figure 11*a*,*b* presents our results for the DOS of phosphorene under a perpendicular magnetic field of 7.94 T. Our simulations are performed for a system with 14 336 × 14 336 unit cells (corresponding to a total of $N = 0.822 \times 10^9$ orbitals). Note that owing to the small energy separation between Landau levels (less than 3 meV), it was necessary to compute a very large number of Chebyshev moments ($M =$ 143 360). The puckered orthorhombic structure of black phosphorus (point group $D_{2h}$) is responsible for its highly anisotropic in-plane electronic properties [110]. To illustrate the anisotropic behaviour of black phosphorus, we use KITE to calculate the DC conductivity tensor (figure 11*c*). As expected, these results show that the mobility of black phosphorus is very sensitive to the direction of in-plane current.

As demonstrated in this example, KITE is extremely advantageous for studies of magneto-transport, since it allows the user to probe the effect of realistic magnetic fields of the order of 1 Tesla with modest computational costs. As a comparison, previous computational studies of black phosphorus were carried out in small lattices (500 × 500 unit cells), limiting the accessible magnetic fields to the range 32.5–130 T [111].

## 4.7. Molecular ensembles

KITE is well suited for the calculation of optical and electronic properties of molecular systems. These are described by a finite set of atomic orbitals, and it is possible to simulate an ensemble of disordered molecules with a single numerical calculation. One approach is to define a molecule as a single unit cell disconnected

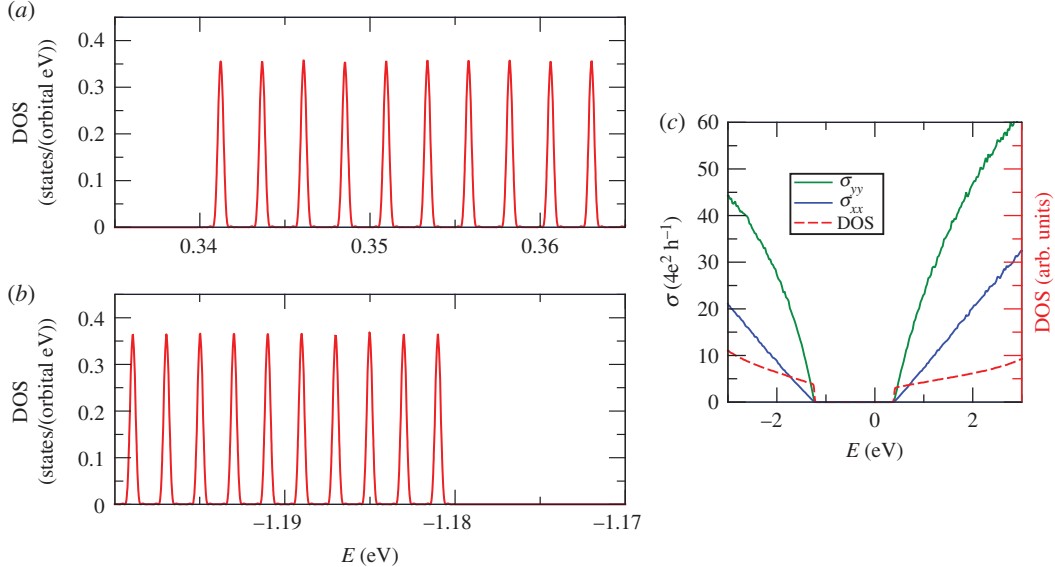

**Figure 11.** (a,b) Average DOS of phosphorene under a perpendicular magnetic field of 7.94 T. Panels (a,b) show the different Landau levels spacing for (a) holes and (b) electrons. Panel (c) shows $\sigma_{xx}(E_F)$ and $\sigma_{yy}(E_F)$ at zero fields, revealing the anisotropy in the DC conductivity of phosphorene. Simulation details: $14\,336 \times 14\,336$ unit cells, $M = 143\,360$ Chebyshev moments and one random vector $R = 1$. Total RAM required for DOS calculation $\approx 39$ GB (double complex precision).

from any neighbouring unit cells. If the user provides an arbitrary set of lattice vectors, a lattice constant larger than the unit cell and lateral sizes, the system is a set of $lx \times ly$ copies of the original molecule. When including any disorder in the unit cell, the system mimics an ensemble of $lx \times ly$ molecules. To illustrate this functionality, we consider a family of oligomers. Oligoacenes are planar molecules consisting of repeated, basic units of a benzene ring [112]. For simplicity, we model the benzene ring as a hexagonal structure with a single $\pi$ orbital per vertex and hopping $t$ between the orbitals. The hoppings within a benzene ring can be specified with the following command

**Listing 15.** Specifying the hopping terms in a benzene ring.

```
lat.add_hoppings(
        ([0, 0], 'C1', 'C2', t),
        ([0, 0], 'C1', 'C3', t),
        ([0, 0], 'C3', 'C5', t),
        ([0, 0], 'C5', 'C6', t),
        ([0, 0], 'C4', 'C6', t),
        ([0, 0], 'C2', 'C4', t)
)
```

It is clear from the above code lines, that adjacent unit cells are disconnected. As such, the script implements isolated benzene molecules. The code lines above can be easily generalized for calculations of any number of benzene rings, always expanding the size of the unit cell to ensure that we are dealing with isolated molecules. This approach can be extended to more complex scenarios, involving multiple orbitals. When including Anderson disorder (see §4.2), the system produces different disorder configurations for each unit cell, which gives rise to a disordered ensemble. Figure 12a shows the density of states of ensembles of disordered oligoacenes with increasing number of rings: benzene, naphthalene, anthracene, tetracene and pentacene. As expected, the size of the central gap ($E = 0$ eV) decreases as a function of the number of rings.

Small oligoacenes, like the ones presented here, can be synthesized on a substrate, in the form of a film. However, due to the fabrication process, the samples can be strongly disordered [113]. To illustrate the convenience of using KITE for molecular analysis, we consider a strongly disordered ensemble of pentacene molecules and calculate their optical conductivity for $E_F = 0$ eV. The results are summarized in figure 12b.

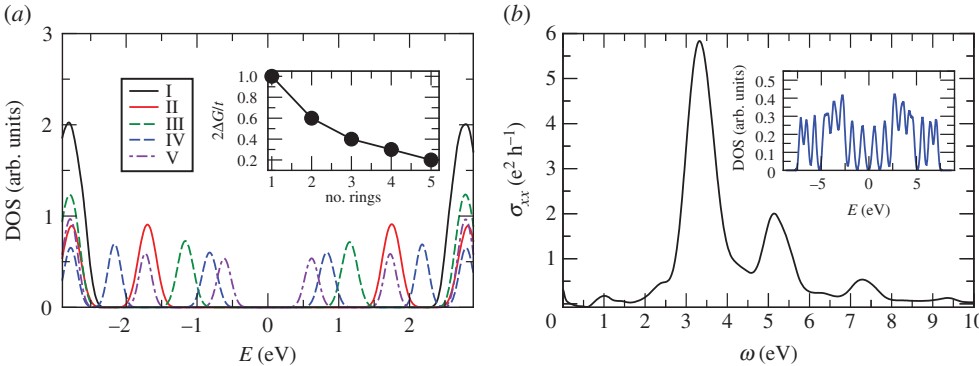

**Figure 12.** (*a*) Density of states of oligoacenes with increasing number of benzene rings. The inset shows the size of the gap as a function of the number of rings. (*b*) Optical conductivity of an ensemble of strongly disordered pentacene molecules. The inset shows the density of states for the sample ensemble. Simulation details: $128 \times 128$ unit cells containing one molecule each and $M = 512$ Chebyshev moments.

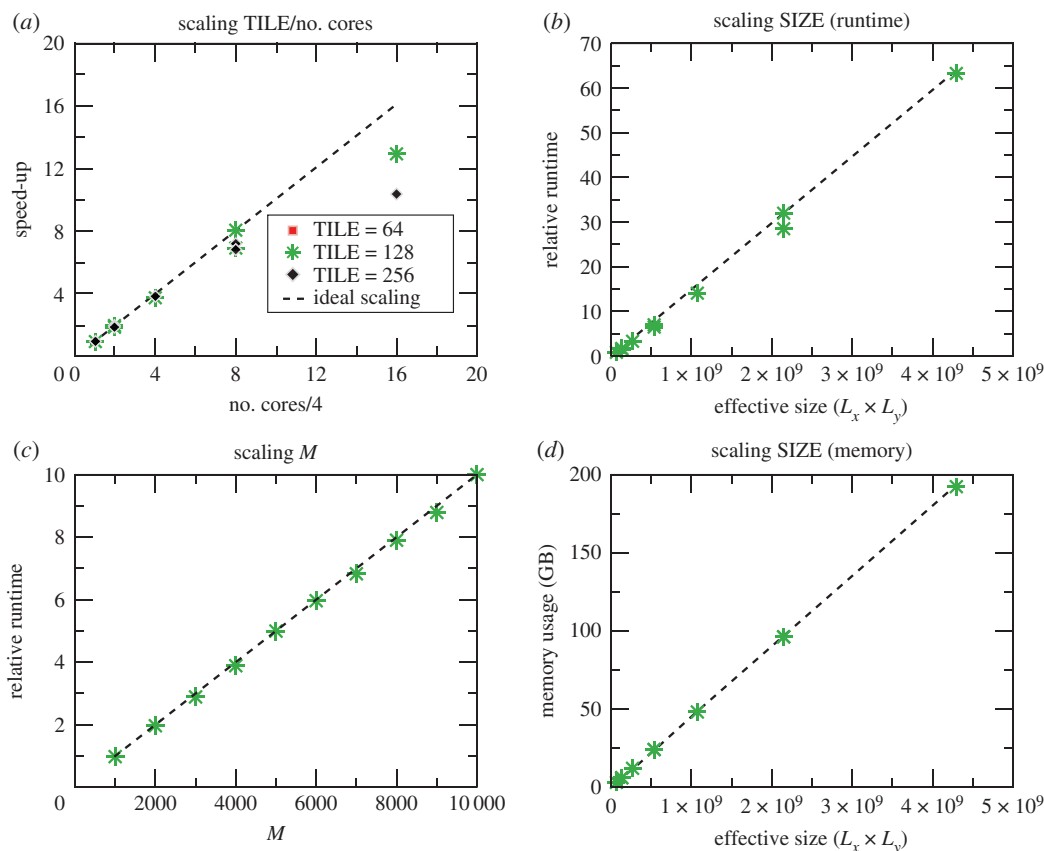

**Figure 13.** Benchmarking the DOS simulations in KITE. (*a*) Speed-up versus number of cores for a given system size, (*b*) relative running time versus effective size of the Hilbert space, (*c*) relative running time versus number of Chebyshev moments, (*d*) memory usage versus effective size of the Hilbert space. In panels (*b*–*d*), 64 cores were used, with a corresponding domain decomposition of 8 $\times$ 8 and TILE $= 64$.

## 5. Benchmarks

Next, we perform a set of simulations to verify how the most critical computational resources scale for different system parameters. First, we benchmark the average DOS for a monolayer of graphene with $Z = 3$. The results are displayed in figure 13 and the numerical values are shown in tables 1–3.

Large-scale simulations on KITE are memory bound, and the calculations involve intensive transfer between the working memory and the processor. Consequently, the efficient use of fast cache memory and an optimized memory transfer can translate in significant speed-ups. This problem is

**Table 1.** Benchmark of the density of states simulation for monolayer graphene with next-nearest hopping, with the system size of $L_1 = L_2 = 16\,384$, coordination number $Z = 3$ and simulation complexity $4.8 \times 10^{12}$, number of moments $M = 1000$ and number of random vectors $R = 1$. The simulations are performed for different values of TILE, and for each TILE, for different number of decomposition domains.

| TILE | Num. Cores | Thr. Distr. | runtime (s) | Rel. Eff. | speed-up |
|---|---|---|---|---|---|
| 64 | 64 | 8 × 8 | 110 | 0.81 | 12.95 |
| 64 | 32 | 8 × 4 | 178 | 1 | 8 |
| 64 | 32 | 4 × 8 | 203 | 0.88 | 7.01 |
| 64 | 16 | 4 × 4 | 378 | 0.94 | 3.77 |
| 64 | 8 | 4 × 2 | 732 | 0.97 | 1.95 |
| 64 | 8 | 2 × 4 | 860 | 0.83 | 1.65 |
| 64 | 4 | 2 × 2 | 1623 | 0.88 | 0.88 |
| 128 | 64 | 8 × 8 | 119 | 0.81 | 12.91 |
| 128 | 32 | 8 × 4 | 192 | 1 | 8 |
| 128 | 32 | 4 × 8 | 221 | 0.87 | 6.95 |
| 128 | 16 | 4 × 4 | 406 | 0.94 | 3.78 |
| 128 | 8 | 4 × 2 | 790 | 0.97 | 1.94 |
| 128 | 8 | 2 × 4 | 815 | 0.94 | 1.88 |
| 128 | 4 | 2 × 2 | 1566 | 0.98 | 0.98 |
| 256 | 64 | 8 × 8 | 153 | 0.65 | 10.38 |
| 256 | 32 | 8 × 4 | 222 | 0.89 | 7.15 |
| 256 | 32 | 4 × 8 | 234 | 0.85 | 6.79 |
| 256 | 16 | 4 × 4 | 411 | 0.97 | 3.86 |
| 256 | 8 | 4 × 2 | 794 | 1 | 2 |
| 256 | 8 | 2 × 4 | 869 | 0.91 | 1.84 |
| 256 | 4 | 2 × 2 | 1620 | 0.98 | 0.98 |

**Table 2.** Benchmark of the density of states simulation for monolayer graphene with next-nearest hopping. TILE is 64, and number of cores is 64 with a domain decomposition of $8 \times 8$. Coordination number $Z = 3$, number of moments $M = 1000$ and number of random vectors is $R = 1$. Different simulations were performed for different system sizes.

| $L_1$ | $L_2$ | Sim. Cplx. | runtime (s) | Rel. Eff. | relative runtime | memory (GB) |
|---|---|---|---|---|---|---|
| 8192 | 8192 | $1.2 \times 10^{12}$ | 31 | 0.85 | 0.85 | 3.10 |
| 8192 | 16 384 | $2.4 \times 10^{12}$ | 63 | 0.86 | 1.72 | 6.12 |
| 16 384 | 8192 | $2.4 \times 10^{12}$ | 57 | 0.78 | 1.56 | 6.12 |
| 16 384 | 16 384 | $4.8 \times 10^{12}$ | 122 | 0.83 | 3.33 | 12.13 |
| 16 384 | 32 768 | $9.7 \times 10^{12}$ | 238 | 0.81 | 6.50 | 24.16 |
| 32 768 | 16 384 | $9.7 \times 10^{12}$ | 257 | 0.88 | 7.02 | 24.16 |
| 32 768 | 32 768 | $1.9 \times 10^{13}$ | 516 | 0.88 | 14.10 | 48.18 |
| 32 768 | 65 536 | $3.9 \times 10^{13}$ | 1044 | 0.89 | 28.53 | 96.25 |
| 65 536 | 32 768 | $3.9 \times 10^{13}$ | 1171 | 1 | 32 | 96.25 |
| 65 536 | 65 536 | $7.7 \times 10^{13}$ | 2316 | 0.99 | 63.29 | 192.29 |

architecture-dependent, and different super-computing machines/clusters offer different speeds of bus lines and different amounts of cache. KITE allows users to tune the code at compilation time and change the rate of transferred data from the main memory. The parameter *TILE* defines the size of the

**Table 3.** Benchmark of the density of states simulation for monolayer graphene with next-nearest hopping. System size is $L_1 = L_2 = 16\,384$, coordination number $Z = 3$ and number of random vectors is $R = 1$. Simulations were performed for different number of moments.

| num. moments | Sim. Cplx. | runtime | Rel. Eff. | relative runtime |
|---|---|---|---|---|
| 1000 | $4.8 \times 10^{12}$ | 115 | 0.98 | 0.98 |
| 2000 | $9.6 \times 10^{12}$ | 231 | 0.98 | 1.97 |
| 3000 | $1.4 \times 10^{13}$ | 339 | 0.96 | 2.89 |
| 4000 | $1.9 \times 10^{13}$ | 456 | 0.97 | 3.89 |
| 5000 | $2.4 \times 10^{13}$ | 586 | 1.00 | 4.99 |
| 6000 | $2.9 \times 10^{13}$ | 700 | 0.99 | 5.97 |
| 7000 | $3.4 \times 10^{13}$ | 801 | 0.98 | 6.83 |
| 8000 | $3.9 \times 10^{13}$ | 927 | 0.99 | 7.90 |
| 9000 | $4.3 \times 10^{13}$ | 1031 | 0.98 | 8.79 |
| 10 000 | $4.8 \times 10^{13}$ | 1173 | 1 | 10 |

simulation subdomain within a single decomposition block that is sent to the processor for calculation. It is important to notice that the optimal value is not only dependent on the size of the system and the amount of cache memory, but also the coordination number of the lattice, precision of the calculation, and chosen decomposition.

To explore further this effect, we give particular attention to the scaling with the number of cores for different TILE values in table 1. A large graphene system with nearest-neighbour hoppings is simulated for three different values of *TILE*: 64, 128 and 256. For each value, we perform a set of simulations for different number of cores (*Num. Cores.*) and different number of decomposition domains along the directions of lattice vectors (*Thr. Distr.*). We define the relative efficiency (*Rel. Eff.*) in the following way: we first take the ratio between the simulation runtime and the runtime of the four cores simulation. This is further scaled by the expected speed-up, given by *Num. Cores*/4. We observe that in some cases, the relative efficiency can become larger than 1. This effect can be explained when considering the different distribution of threads, done both by the user and by the operating system on the computational machine. Namely, datasets in KITE are aligned along the directions of the first lattice vectors. For example, in memory, the neighbouring unit cells are [0, 0], [1, 0], [2, 0], etc., and it is more efficient to have more decomposition domains along this direction. The second effect is the distribution of threads when a small number of cores is chosen. All the threads are bound to cores where they were initially spawned using the *OpenMP* OMP_PROC_BIND flag, but their initial distribution depends on the operating system. Due to these uncontrollable effects small variations in the relative efficiency arise. We therefore further rescale the relative efficiency with the maximum one. This ensures that the maximal relative efficiency is fixed to 1, which, although it decreases the efficiency of some simulations, ensures that the reference is solely defined by the 'most efficient' simulation.

We further define the speed-up (*Speedup*) as the product between the relative efficiency and the expected speed-up. As shown in figure 13a, the speed-up is almost linear with a noticeable saturation effect for large *TILE* of 256. We use this parameter to define the optimal value of 64 (or almost equally efficient 128), which is the value we use further in this section.

We also benchmark the relative running time (*Relative runtime*), taking into account the previous discussion on the efficiency, and the used RAM memory (*Memory (GB)*) with respect to the system size, or the simulation complexity (*Sim. Cplx.*), which is a measure of the number of the operations that have to be performed, and it is equal to the product between the system size ($L_1$ and $L_2$), number of orbitals within the unit cell ($n_O$), coordination number $Z$, number of expansion moments $M$ and number of random vectors ($R$). The results are presented in table 2 and plotted in figure 13b,d. One can see that the scaling is excellent, i.e. close to the theoretical $O(Sim.\ Cplx.)$ scaling, for all system sizes. Regarding the memory requirements, the scaling is perfectly linear, which further proves that memory usage is highly optimized.

We finally present the benchmark of the DOS with respect to the number of moments in the CPGF expansion ($M$). The results, presented in table 3 and plotted in figure 13c, demonstrate that the relative

running time follows the trend of perfect scaling when increasing the number of moments. From this, we can conclude that KITE is well optimized; most of the simulation time is spent on the calculation of the Chebyshev moments. This is also important for estimating the runtime of other simulations in KITE. The conductivity, for example, generally needs a double-expansion, therefore $M^2$ moments are needed. In this case, the simulation scales quadratically with $M$.

In addition to the optimization that KITE offers, users are encouraged to test the hardware and fine tune TILE and other parameters to their needs. Different topologies determined by the lattice connectivity results in different performances, and we firmly suggest that for the best performance, one should first perform a similar benchmarking test.

The results in this section were obtained using KITE v. 1.0, compiled with gcc 7.3. All the simulations were run on high memory nodes of the VIKING cluster, the York Advanced Research Computing Cluster.

# 6. Conclusion

KITE is an open-source software suite for accurate electronic structure and quantum transport calculations in real-space based on Chebyshev spectral expansions of lattice Green's functions. It features a Python-based interface that allows for an intuitive simulation set-up and workflow automation of tight-binding calculations, with a wide range of pre-defined observables to assess electronic structure and non-equilibrium properties of complex molecular and condensed systems in the presence of disorder, defects and external stimuli. However, the main advantage of KITE is the way the software is written, being ideal for *user-friendly quantum transport calculations*, at the same time allowing unprecedented simulations of *realistic structures containing multi-billions of orbitals*. The user can optimize the calculations, workflow and post-processing at various levels on standard and high-performance computing machines to best maximize resources. The computational cost is divided physically among the processor units by means of a lattice domain decomposition technique for efficient evaluation of Chebyshev recursions on large Hilbert spaces. Combined with an efficient memory management scheme, this strategy enables KITE to achieve nearly linear computational speed-ups with increasing numbers of processors, to our knowledge not previously demonstrated in a tight-binding code. We have also demonstrated linear scaling in the multi-threading performance with respect to the number of Chebyshev polynomials and size of computational domain (Hilbert space), showing that KITE is capable to take full advantage of available computational resources in a variety of scenarios. Through the provided examples we showed that KITE can easily outperform previous state-of-the-art tight-binding simulations. For example, the DOS calculation for low-angle twisted bilayer graphene, using a tight-binding model with high coordination number $Z = 60$ (§4.1), employed 3.9 billion orbitals, which is two to three orders of magnitude larger than typical sizes used in studies of simple (sparse) graphene lattices ($Z = 3$), using either the kernel polynomial method [38] or hybrid spectral approaches [37]. It also overturns previous world-record tight-binding calculations with 3.6 billion orbitals [43], where the Chebyshev polynomial Green's function methodology, at the heart of KITE, was originally proposed. The major difference with respect to the standard OPENMP parallelization in [43] lies in the superior memory management of KITE, which allows to simulate ever larger and more complex tight-binding models. Another significant advance is demonstrated in the calculation of the conductivity tensor of disordered topological materials at finite temperature where 268 million orbitals were used, far above the previous reports in the literature for systems with equivalent complexity (for example, 16 million orbitals in a similar calculation in [114]). KITE can handle even larger systems provided enough RAM memory is available. Since the memory usage is proportional to the system size, calculations of systems with $N = O(10^{10})$ and beyond can be run on high-memory nodes ($\geq$256 GB) already available in most computing clusters.

KITE has a sophisticated, yet flexible way of introducing crystalline defects and disorder, which preserves the efficiency of the underlying Chebyshev recursion schemes. When combined with its efficient domain decomposition algorithm allowing accurate and fast calculations with billions of orbitals, this capability makes KITE particularly suited for investigations of the effect of disorder and external perturbations on two-dimensional materials and their heterostructures, thin films and multi-layer structures.

With its first release, KITE already boasts a generous range of functionalities, such as local density of states, spectral functions, wave-packet propagation, longitudinal and transverse DC and optical conductivity, and nonlinear optical conductivity. Because of its modular design and open-source nature, it is possible to implement new features in KITE efficiently, and the authors hope that new functionalities can be constructed by users and groups interested in interfacing their codes with KITE. Further developments and extensions of KITE will be driven by the needs of the communities actively

using it. To lower the entry level for researchers to use and help developing KITE, we intend to use KITE website as a platform for providing extensive tutorials with hands-on sessions using KITE.

Data accessibility. KITE's webpage contains updated versions of KITE's source, installation instructions and tutorials (http://quantum-kite.com) [20] KITE's source, examples and the configuration files used to generate the data of this article are also available in Zenodo (https://doi.org/10.5281/zenodo.3485089) [70].

Authors' contributions. A.F. and T.G.R. conceived the project. J.M.V.P.L. directed the code development. L.C. directed the interface development. A.F. conceived Green's function spectral methodology and supervised the algorithmic design. J.M.V.P.L. conceived and implemented the template design of operators and the multi-scale domain composition algorithms of KITE. M.A. wrote the Python-based interface. S.M.J. wrote the post-processing tools. S.M.J. and J.M.V.P.L. wrote the core code and designed and implemented the automated Peierls-phase functionality. All authors contributed to the research leading to the examples demonstrating KITE functionalities and analysed the results. A.F., T.G.R., J.V.M.P.L. and L.C. wrote the manuscript and software documentation with input from M.A. and S.M.J.

Competing interests. We declare we have no competing interest.

Funding. T.G.R. and A.F. acknowledge support from the Newton Fund and the Royal Society through the Newton Advanced Fellowship scheme (ref. no. NA150043). M.A. and L.C. acknowledge support from the Trans2DTMD FlagEra project and the VSC (Flemish Supercomputer Center). A.F. acknowledges support from the Royal Society through a University Research Fellowship (ref. nos. UF130385 and URF-R-191021) and an Enhancement Award (ref. no. RGF-EA-180276). T.G.R. acknowledges the support from the Brazilian agencies CNPq and FAPERJ and COMPETE2020, PORTUGAL2020, FEDER and the Portuguese Foundation for Science and Technology (FCT) through project POCI-01- 0145-FEDER-028114. S.M.J. is supported by Fundação para a Ciência e Tecnologia (FCT) under the grant no. PD/BD/142798/2018. S.M.J. and J.M.V.P.L. acknowledge financial support from the FCT, COMPETE 2020 programme in FEDER component (European Union), through projects POCI-01-0145-FEDER-028887 and UID/FIS/04650/2013. S.M.J. and J.M.V.P.L. further acknowledge financial support from FCT through national funds, co-financed by COMPETE-FEDER (grant no. M-ERANET2/0002/2016 – UltraGraf) under the Partnership Agreement PT2020.

Acknowledgements. We are grateful to the Royal Society for supporting the inception of KITE open-source initiative. We acknowledge fruitful discussions and encouragement by B. Amorim, A. Pachoud, M. D. Costa, N. M. R. Peres, E. R. Mucciolo, P. Hasnip and S. Roche. We thank technical support from M. D. Costa and K. Murphy (high-performance computing) and J. Giannella (web design). This project was undertaken on the Viking Cluster, which is a high-performance computer facility provided by the University of York. We are grateful for computational support from the University of York High Performance Computing service, Viking and the Research Computing team.

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
