## [Reviewer comments · Royal Society Open Science]

Review History

RSOS-191809.R0 (Original submission)

Review form: Reviewer 1

Is the manuscript scientifically sound in its present form?

Yes

Are the interpretations and conclusions justified by the results?

Yes

Is the language acceptable?

Yes

Do you have any ethical concerns with this paper?

No

Have you any concerns about statistical analyses in this paper?

No

Recommendation?

Accept with minor revision (please list in comments)

Comments to the Author(s)

The present paper titled “KITE: high-performance accurate modelling of electronic structure and response functions of large molecules, disordered crystals and heterostructures” presents the theoretical basis, key examples, and benchmarks of the authors newly developed quantum transport code KITE. Overall I think this is a very nice piece of work and will provide a useful tool in the ever expanding field of large scale tight-binding simulations. Below I give some comments on the work.

** On page 15 of the manuscript, the authors show an example for non-relaxed and relaxed tBLG, it is not clear if the MD relaxation was performed within KITE or not? If yes, can other potential schemes be used, e.g., Tersoff. It would certainly be more realistic to allow for the relaxation such large heterostructures, especially when considering the local environment surrounding defects and in other forms of disorder.

** In connection to the last comment, are there any links to first-principles codes for constructing or importing (Wannier90 type) tight-binding models? I looked through the documentation of Pybinding, but I could not find any reference to such methods. Since this is a very efficient code for examining large scale systems, it would be very useful (and straight forward) to be able to use tight-binding parameterizations from ab initio results to enhance comparison to experiments and in predicting new robust devices.

** The authors describe a very simple form of disorder (Page 16), where the onsite energies are chosen to fluctuate randomly. Since, the abstract of the paper implies to claim the code can handle disordered materials more broadly, can other approaches to disorder be executed, e.g., VCA, CPA, etc..? I agree that the framework can attack any tight-binding Hamiltonian, but by claiming to capture disordered systems I would expect a more sophisticated approach. Now, the authors may have meant that they can capture disorder by forming large real space system with random arrangements of defects, atomic substitutions, etc..., this I agree with, and I would stress this more in the paper, since the only example of disorder given uses the simple energetic treatment. Do the authors have an example with real disorder to illustrate the power of the method? Perhaps defects in magic angle tBLG?

** On page 20 and 21 the authors show the spectral function and its connection to the real space basis. I might have missed something, but why was this approach taken when the Green's function is already calculated? Is it much more expensive to Fourier transfer the GF? Also, it is not clear if the spectral function can be projected on to various orbitals or sites. To compare to experiment, various components of the spectral weight might be of key importance in systems with broken symmetries such as AMF order.

Furthermore, the manuscript claims to provide simulation of photoemission spectroscopy, however I strongly disagree. I do not find KITE to calculate the single or 3-step ARPES matrix element as discussed in Phys. Rev. B 65, 054514 (2002) or Phys. Rev. Lett. 83, 5154 (1999). If the authors only mean that they are able to calculate the spectral functions as a proxy of the ARPES spectra I would make this claim instead and not state they calculate the observed spectra. It might be common to state that the spectral function is the ARPES spectra, but they can give very different results depending on the experimental setup, e.g., termination, surface morphology, photon energy, and so on. I would suggest that the authors take the approach as they did in presenting the LDOS. Specifically, I would state that spectral functions can be calculated and say the spectral function can be related to the ARPES spectra, just as the LDOS is related to STM/STS spectra but it is not the observed spectra.

** Do the authors see any future porting to GPUs to allow them to attack even larger systems?

Minor comments:

Typo Eq. 2.7 $T_0(X) = 1$

In closing, this is a nice set of tools for attacking these class challenging problems and once my comments have been addressed, I see no reason not to recommend this work for publication.

Review form: Reviewer 2

Is the manuscript scientifically sound in its present form?

Yes

Are the interpretations and conclusions justified by the results?

Yes

Is the language acceptable?

Yes

Do you have any ethical concerns with this paper?

No

Have you any concerns about statistical analyses in this paper?

No

Recommendation?

Accept with minor revision (please list in comments)

Comments to the Author(s)

The paper "KITE: high performance accurate modelling of electronic structure and response functions of large molecules, disordered crystals and heterostructures" by Simao et al. introduces a new modelling software that is capable to provide insights into the properties of extremely large systems. The work is based on the tight-binding approximation and use a combination of advanced numerical techniques to diagonalize the resulting Hamiltonian and compute various properties, including conductivities, optical properties, local density of states, etc. The paper gives a detailed description of the formalism, presents the layout of the code and showcases a number of applications to interesting systems. Overall, I believe this is an excellent paper that deserves publications after a few comments (see below) are addressed.

+ it is stated that the empirical tight-binding method is used, but it is never explicitly explained what this means, ie what does the word "empirical" mean in this context? This should be clarified in the introduction without appropriate references.

+ in my opinion, the authors oversell the power of the tight-binding method. In my own experience, it can be quite tricky to generate accurate and transferable tight-binding representations for chemically complex systems, in particular if it wants to avoid long-ranged hoppings. For example, it is well known that simple tight-binding methods can yield inaccurate (and even qualitatively wrong) results for the properties of defects in materials (e.g. if one would simply remove an atom from the tight-binding Hamiltonian for a TMD monolayer, this would give different results than a full DFT simulation). I believe the authors should be more cautious and warn the readers about the limitations of the tight-binding method.

+ another key limitation of the TB method is that it describes non-interacting electrons. This means that if the electron density is inhomogeneous, as in twisted bilayer graphene, the tight-binding method will not capture the electric field generated by the electron puddles which in turn makes the charge density more uniform than what TB would predict. Another field where long-ranged electron-electron interactions are important are optical properties where they give rise to the formation of plasmons and excitons. Again, it is important to caution the reader that the presented formalism cannot capture the important effects.

Decision letter (RSOS-191809.R0)

16-Dec-2019

Dear Dr Rappoport

On behalf of the Editors, I am pleased to inform you that your Manuscript RSOS-191809 entitled "KITE: high-performance accurate modelling of electronic structure and response functions of large molecules, disordered crystals and heterostructures" has been accepted for publication in Royal Society Open Science subject to minor revision in accordance with the referee suggestions. Please find the referees' comments at the end of this email.

The reviewers and handling editors have recommended publication, but also suggest some minor revisions to your manuscript. Therefore, I invite you to respond to the comments and revise your manuscript.

- Ethics statement

- Data accessibility

<http://datadryad.org/submit?journalID=RSOS&manu=RSOS-191809>

- Competing interests

- Authors' contributions

All submissions, other than those with a single author, must include an Authors' Contributions section which individually lists the specific contribution of each author. The list of Authors

should meet all of the following criteria; 1) substantial contributions to conception and design, or acquisition of data, or analysis and interpretation of data; 2) drafting the article or revising it critically for important intellectual content; and 3) final approval of the version to be published.

- Acknowledgements

- Funding statement

Because the schedule for publication is very tight, it is a condition of publication that you submit the revised version of your manuscript before 25-Dec-2019. Please note that the revision deadline will expire at 00.00am on this date. If you do not think you will be able to meet this date please let me know immediately.

- 1) A text file of the manuscript (tex, txt, rtf, docx or doc), references, tables (including captions) and figure captions. Do not upload a PDF as your "Main Document";
- 2) A separate electronic file of each figure (EPS or print-quality PDF preferred (either format should be produced directly from original creation package), or original software format);

- 3) Included a 100 word media summary of your paper when requested at submission. Please ensure you have entered correct contact details (email, institution and telephone) in your user account;
- 4) Included the raw data to support the claims made in your paper. You can either include your data as electronic supplementary material or upload to a repository and include the relevant doi within your manuscript. Make sure it is clear in your data accessibility statement how the data can be accessed;
- 5) All supplementary materials accompanying an accepted article will be treated as in their final form. Note that the Royal Society will neither edit nor typeset supplementary material and it will be hosted as provided. Please ensure that the supplementary material includes the paper details where possible (authors, article title, journal name).

If your manuscript is newly submitted and subsequently accepted for publication, you will be asked to pay the article processing charge, unless you request a waiver and this is approved by Royal Society Publishing. You can find out more about the charges at <https://royalsocietypublishing.org/rsos/charges>. Should you have any queries, please contact openscience@royalsociety.org.

on behalf of Dr Robert Young (Associate Editor) and Miles Padgett (Subject Editor)
openscience@royalsociety.org

Associate Editor Comments to Author (Dr Robert Young):

Associate Editor: 1

Comments to the Author:

Please respond to the comments from the two reviewers carefully.

Reviewer comments to Author:

Reviewer: 1

Comments to the Author(s)

The present paper titled "KITE: high-performance accurate modelling of electronic structure and response functions of large molecules, disordered crystals and heterostructures" presents the

theoretical basis, key examples, and benchmarks of the authors newly developed quantum transport code KITE. Overall I think this is a very nice piece of work and will provide a useful tool in the ever expanding field of large scale tight-binding simulations. Below I give some comments on the work.

** On page 15 of the manuscript, the authors show an example for non-relaxed and relaxed tBLG, it is not clear if the MD relaxation was performed within KITE or not? If yes, can other potential schemes be used, e.g., Tersoff. It would certainly be more realistic to allow for the relaxation such large heterostructures, especially when considering the local environment surrounding defects and in other forms of disorder.

** In connection to the last comment, are there any links to first-principles codes for constructing or importing (Wannier90 type) tight-binding models? I looked through the documentation of Pybinding, but I could not find any reference to such methods. Since this is a very efficient code for examining large scale systems, it would be very useful (and straight forward) to be able to use tight-binding parameterizations from ab initio results to enhance comparison to experiments and in predicting new robust devices.

** The authors describe a very simple form of disorder (Page 16), where the onsite energies are chosen to fluctuate randomly. Since, the abstract of the paper implies to claim the code can handle disordered materials more broadly, can other approaches to disorder be executed, e.g., VCA, CPA, etc.? I agree that the framework can attack any tight-binding Hamiltonian, but by claiming to capture disordered systems I would expect a more sophisticated approach. Now, the authors may have meant that they can capture disorder by forming large real space system with random arrangements of defects, atomic substitutions, etc..., this I agree with, and I would stress this more in the paper, since the only example of disorder given uses the simple energetic treatment. Do the authors have an example with real disorder to illustrate the power of the method? Perhaps defects in magic angle tBLG?

** On page 20 and 21 the authors show the spectral function and its connection to the real space basis. I might have missed something, but why was this approach taken when the Green's function is already calculated? Is it much more expensive to Fourier transfer the GF? Also, it is not clear if the spectral function can be projected on to various orbitals or sites. To compare to experiment, various components of the spectral weight might be of key importance in systems with broken symmetries such as AMF order.

Furthermore, the manuscript claims to provide simulation of photoemission spectroscopy, however I strongly disagree. I do not find KITE to calculate the single or 3-step ARPES matrix element as discussed in Phys. Rev. B 65, 054514 (2002) or Phys. Rev. Lett. 83, 5154 (1999). If the authors only mean that they are able to calculate the spectral functions as a proxy of the ARPES spectra I would make this claim instead and not state they calculate the observed spectra. It might be common to state that the spectral function is the ARPES spectra, but they can give very different results depending on the experimental setup, e.g., termination, surface morphology, photon energy, and so on. I would suggest that the authors take the approach as they did in presenting the LDOS. Specifically, I would state that spectral functions can be calculated and say the spectral function can be related to the ARPES spectra, just as the LDOS is related to STM/STS spectra but it is not the observed spectra.

** Do the authors see any future porting to GPUs to allow them to attack even larger systems?

Minor comments:

Typo Eq. 2.7 $T_0(X) = 1$

In closing, this is a nice set of tools for attacking these class challenging problems and once my comments have been addressed, I see no reason not to recommend this work for publication.

Reviewer: 2

Comments to the Author(s)

The paper "KITE: high performance accurate modelling of electronic structure and response functions of large molecules, disordered crystals and heterostructures" by Simao et al. introduces a new modelling software that is capable to provide insights into the properties of extremely large systems. The work is based on the tight-binding approximation and use a combination of advanced numerical techniques to diagonalize the resulting Hamiltonian and compute various properties, including conductivities, optical properties, local density of states, etc. The paper gives a detailed description of the formalism, presents the layout of the code and showcases a number of applications to interesting systems. Overall, I believe this is an excellent paper that deserves publications after a few comments (see below) are addressed.

+ it is stated that the empirical tight-binding method is used, but it is never explicitly explained what this means, ie what does the word "empirical" mean in this context? This should be clarified in the introduction without appropriate references.

+ in my opinion, the authors oversell the power of the tight-binding method. In my own experience, it can be quite tricky to generate accurate and transferable tight-binding representations for chemically complex systems, in particular if it wants to avoid long-ranged hoppings. For example, it is well known that simple tight-binding methods can yield inaccurate (and even qualitatively wrong) results for the properties of defects in materials (e.g. if one would simply remove an atom from the tight-binding Hamiltonian for a TMD monolayer, this would give different results than a full DFT simulation). I believe the authors should be more cautious and warn the readers about the limitations of the tight-binding method.

+ another key limitation of the TB method is that it describes non-interacting electrons. This means that if the electron density is inhomogeneous, as in twisted bilayer graphene, the tight-binding method will not capture the electric field generate by the electron puddles which in turn makes the charge density more uniform than what TB would predict. Another field where long-ranged electron-electron interactions are important are optical properties where they give rise to the formation of plasmons and excitons. Again, it is important to caution the reader that the presented formalism cannot capture the important effects.

Author's Response to Decision Letter for (RSOS-191809.R0)

See Appendix A.

Decision letter (RSOS-191809.R1)

17-Jan-2020

Dear Dr Rappoport,

It is a pleasure to accept your manuscript entitled "KITE: high-performance accurate modelling of electronic structure and response functions of large molecules, disordered crystals and heterostructures" in its current form for publication in Royal Society Open Science. The comments of the reviewer(s) who reviewed your manuscript are included at the foot of this letter.

on behalf of Dr Robert Young (Associate Editor) and Miles Padgett (Subject Editor)
openscience@royalsociety.org

Appendix A

Resubmission of RSOS-191809

Dear Editors,

Firstly, we would like to thank the editors of Royal Society Open Science for arranging the review of our manuscript. Secondly, we are thankful to the Referees for their constructive criticism, which is now reflected in the revised manuscript. Referee 1 states: "*Overall I think this is a very nice piece of work and will provide a useful tool in the ever expanding field of large scale tight-binding simulations*" while referee 2 states: "*The paper gives a detailed description of the formalism, presents the layout of the code and showcases a number of applications to interesting systems. Overall, I believe this is an excellent paper that deserves publications after a few comments (see below) are addressed.*"

We feel that with the improvements suggested by both referees, the revised manuscript is now ready for publication in the Royal Society Open Science journal.

In what follows, we address the comments and questions of both referees and provide a summary of changes made, as requested.

Yours sincerely,

Tatiana G. Rappoport,

on behalf of the authors

S. M. João, M. Anđelković, L. Covaci, T. G. Rappoport, J. M. V. P. Lopes and A. Ferreira

Reply to Referee 1

On page 15 of the manuscript, the authors show an example for non-relaxed and relaxed tBLG, it is not clear if the MD relaxation was performed within KITE or not? If yes, can other potential schemes be used, e.g., Tersoff. It would certainly be more realistic to allow for the relaxation such large heterostructures, especially when considering the local environment surrounding defects and in other forms of disorder.

The molecular dynamics simulations were performed in a separated software (LAMMPS) by considering a combination of Brenner potentials for in-plane interactions and a registry-dependent Kolmogorov-Crespi potential. We have now clarified this relevant point in the manuscript.

In connection to the last comment, are there any links to first-principles codes for constructing or importing (Wannier90 type) tight-binding models? I Looked through the documentation of Pybinding, but I could not find any reference to such methods. [...]

Yes, there is a script on PythTB written by Vanderbilt and co-workers that imports tight-binding parameters directly from Wannier90 (see <http://physics.rutgers.edu/pythtb/examples.html?highlight=wan>). We cite PythTb in the article.

[...] Since this is a very efficient code for examining large scale systems, it would be very useful (and straight forward) to be able to use tight-binding parameterizations from ab initio results to enhance comparison to experiments and in predicting new robust devices.

We agree that such a functionality would be very useful. We are currently developing a dedicated function to import the output of Wannier90 into KITE. We hope it will be available in version 2 of KITE.

The authors describe a very simple form of disorder (Page 16), where the onsite energies are chosen to fluctuate randomly. Since, the abstract of the paper implies to claim the code can handle disordered materials more broadly, can other approaches to disorder be executed, e.g., VCA, CPA, etc..? I agree that the framework can attack any tight-binding Hamiltonian, but by claiming to capture disordered systems I would expect a more sophisticated approach. Now, the authors may have meant that they can capture disorder by forming large real space system with random arrangements of defects, atomic substitutions, etc. . . , this I agree with, and I would stress this more in the paper, since the only example of disorder given uses the simple energetic treatment. Do the authors have an example with real disorder to illustrate the power of the method? Perhaps defects in magic angle tBLG?

The different types of disorder and their implementations, including detailed examples, can be found on KITE's website. The disorder functionality on KITE (one of its major highlights) makes use of an intuitive real space "disorder cell" approach, which allows to set up virtually any modification of the original tight-binding lattice, which is then treated exactly by KITE's spectral algorithms. This encompasses not only the examples in the article (random on-site energies (Secs. 4b-c, 4d(i), 4f-g), local Ising moments (4e) and vacancies (4d(ii))), but many other disorders, including bond disorder, random gauge fields and generic multi-orbital disorder, just to mention a few examples. The article is

focused on showcasing the large-scale capabilities of the software (allowing tight-binding simulations of spectral/response functions with unprecedented energy resolution), rather than illustrating the many types of disorder that can be efficiently handled by KITE. We emphasize that the implemented spectral approach is fully nonperturbative, offering control over energy resolution, and is therefore superior to any of the conventional approximate schemes for static disorder, such as VCA, single-impurity T-matrix or CPA. KITE allows the evaluation of generalised spectral/response functions for arbitrary 2D/3D lattice with disorder effects fully accounted for. Indeed, within the CPGF method, the full set of electron-impurity scattering events are effectively "summed up", within the desired energy resolution and numerical accuracy. This opens up the interesting prospect of using KITE to test and improve conventional schemes for calculations of self energies, vertex corrections to response functions, etc. Notwithstanding, we agree with the Referee that with a large number of examples, it could be instructive to illustrate the implementation of more complex disorder. With this in mind, we included a new example in Section 4e: random spin-orbit-active ad-atoms located at the center of hexagonal plaquettes (hollow site). This is a complex local disorder, which gives rise to 5 types of interactions: Rashba interaction, intrinsic-type spin-orbit coupling, scalar potential scattering, third-neighbor hopping and chiral spin-dependent nearest-neighbor hoppings around the plaquette. The example shows how the spin dynamics can be profoundly altered by the presence of the heavy impurities, illustrating the impact of realistic (complex) disorder on an experimentally relevant transport quantity.

On page 20 and 21 the authors show the spectral function and its connection to the real space basis. I might have missed something, but why was this approach taken when the Green's function is already calculated? Is it much more expensive to Fourier transfer the GF?

KITE's approach consists of evaluating traces and matrix elements of generic operators (in real-space representation) by means of an exact Chebyshev decomposition of the Green's function (GF) [Eq. (2.19)] or the resolvent operator [Eq. (2.12)]. Taking as an example the spectral function alluded by the Referee, it is more advantageous to compute the matrix overlap [Eq. (4.5)] for a grid of k points, than to evaluate the individual $N \times N$ complex entries of the GF matrix (where N can be as large as 10^{10}) and subsequently perform a Fourier transform.

Also, it is not clear if the spectral function can be projected on to various orbitals or sites. To compare to experiment, various components of the spectral weight might be of key importance in systems with broken symmetries such as AMF order.

The idea behind calculating the spectral functions is the possibility of comparing ARPES spectra of disordered systems with real-space calculations. In this first version of KITE, the projection on to various orbitals is not implemented yet. However, this is a straightforward extension to our software, which will be included in our next update, together with generalised conductivities and spin/orbital projections of the density of states.

Furthermore, the manuscript claims to provide simulation of photoemission spectroscopy, however I strongly disagree. I do not find KITE to calculate the single or 3-step ARPES matrix element as discussed in Phys. Rev. B 65, 054514 (2002) or Phys. Rev. Lett. 83, 5154 (1999). If the authors only mean that they are able to calculate the spectral functions as a proxy of the ARPES spectra I would make this claim instead and not

state they calculate the observed spectra. It might be common to state that the spectral function is the ARPES spectra, but they can give very different results depending on the experimental setup, e.g., termination, surface morphology, photon energy, and so on. I would suggest that the authors take the approach as they did in presenting the LDOS. Specifically, I would state that spectral functions can be calculated and say the spectral function can be related to the ARPES spectra, just as the LDOS is related to STM/STS spectra but it is not the observed spectra.

We agree with the referee. As aforementioned, although the calculation of ARPES matrix elements within KITE offers no difficulty, this is not yet implemented in the current version. We have revised the text accordingly.

Do the authors see any future porting to GPUs to allow them to attack even larger systems?

The KITE internal structure is compatible with both GPU and MPI-based parallel programming. At the moment, we are working in close collaboration with York Research Computing to implement MPI. We prioritised OpenMP due the memory requirements. KITE was designed to achieve the largest possible system sizes (to maximise energy resolution). A typical simulation uses dozens or hundreds of GBs. When we started the development of KITE, the memory available on our GPU computers was very limited. Nowadays, there are cards with 32 GB (with a cost around GBP 8K), which could be and interesting option to allocate each lattice domain of a domain decomposition. It is nevertheless an expensive architecture, but we think that it could allow an interesting increase of performance.

Minor comments: Typo Eq. 2.7 $T0(X) = 1$

We thank the referee for identifying the typo in Eq. (2.7).

Reply to Referee 2

it is stated that the empirical tight-binding method is used, but it is never explicitly explained what this means, ie what does the word "empirical" mean in this context? This should be clarified in the introduction without appropriate references.

The expression ("empirical"/"semi-empirical" TB as opposed to first-principles TB) is commonly used in quantum chemistry and condensed matter physics, but we understand it should be clarified for a more general audience. We re-wrote part of the introduction to be more general and cross-disciplinary and included a citation to the review article "Ab initio tight binding" by A P Horsfield-dag and A M Bratkovsky, which illustrates well the limitations of the empirical tight-binding (TB) approach.

in my opinion, the authors oversell the power of the tight-binding method. In my own experience, it can be quite tricky to generate accurate and transferable tight-binding representations for chemically complex systems, in particular if it wants to avoid long-ranged hoppings. For example, it is well known that simple tight-binding methods can yield inaccurate (and even qualitatively wrong) results for the properties of defects in materials (e.g. if one would simply remove an atom from the

tight-binding Hamiltonian for a TMD monolayer, this would give different results than a full DFT simulation). I believe the authors should be more cautious and warn the readers about the limitations of the tight-binding method.

We fully agree with the referee about the difficulty of constructing accurate and transferable tight-binding Hamiltonians. This is exactly why we constructed a software where it is technically simple to work with multiple orbitals that can be parametrised from first-principle calculations. KITE can deal with arbitrary complex tight-binding models (with orthogonal basis set) with long-range hoppings in a very general way. Defects/impurities can be modelled by sophisticated structures that can be parametrised by means of dedicated first-principle calculations. With these features, we believe that KITE can deliver realistic and large scale calculations of complex molecules and condensed systems. Nevertheless, we have revised the Introduction, to make it clear the limitations and difficulties associated with the tight-binding method.

another key limitation of the TB method is that it describes non-interacting electrons. This means that if the electron density is inhomogeneous, as in twisted bilayer graphene, the tight-binding method will not capture the electric field generate by the electron puddles which in turn makes the charge density more uniform than what TB would predict. Another field where long-ranged electron-electron interactions are important are optical properties where they give rise to the formation of plasmons and excitons. Again, it is important to caution the reader that the presented formalism cannot capture the important effects.

We agree with the referee and give more emphasis to the limitations of the tight-binding formalism.

Summary of changes made:

1. The main changes in the manuscript that are made to improve the manuscript to comply with the points raised by the referees are presented in red in the PDF file. However, we revised the text and made several grammar edits that are not marked.
2. Following the suggestions of referee 2, we re-wrote the first two pages of the introduction to be more general, cross-disciplinary, and emphasize the limitations of the tight-binding formalism.
3. On page 16, we included a sentence and references to explain the relaxation process for the twisted bilayer, according to the request of referee 1.
4. On page 21, we removed any mention to ARPES calculations and used spectral function instead.
5. On pages 24 and 26, in subsection *Spintronics:time-evolution of spin-polarized wave-packets*, we included a new example of random spin-orbit-active ad-atoms located at the center of hexagonal plaquettes (hollow site), a case of complex disorder, that was missing in the article. A new panel in Figure 10 shows the spin-relaxation for this type of disorder and we included a new paragraph on page 26 to discuss the new results. Due to space limitations and the size of the script, we do not present the plaquette disorder in detail in the article. Still, the new script is now part of the available open data that accompanies our work.